# Monosymmetric Fe-N₄ sites enabling durable proton exchange membrane fuel cell cathode by chemical vapor modification

Jingsen Bai[1,2,7], Tuo Zhao[3,7], Mingjun Xu[1,2,7], Bingbao Mei[4,7], Liting Yang[1,2], Zhaoping Shi[1,2], Siyuan Zhu [1,2], Ying Wang [5] ✉, Zheng Jiang [6], Jin Zhao [1,2] ✉, Junjie Ge [1,2] ✉, Meiling Xiao[1,2], Changpeng Liu[1,2] & Wei Xing [1,2] ✉

The limited durability of metal-nitrogen-carbon electrocatalysts severely restricts their applicability for the oxygen reduction reaction in proton exchange membrane fuel cells. In this study, we employ the chemical vapor modification method to alter the configuration of active sites from FeN₄ to the stable monosymmetric FeN₂+N'₂, along with enhancing the degree of graphitization in the carbon substrate. This improvement effectively addresses the challenges associated with Fe active center leaching caused by N-group protonation and free radicals attack due to the 2-electron oxygen reduction reaction. The electrocatalyst with neoteric active site exhibited excellent durability. During accelerated aging test, the electrocatalyst exhibited negligible decline in its half-wave potential even after undergoing 200,000 potential cycles. Furthermore, when subjected to operational conditions representative of fuel cell systems, the electrocatalyst displayed remarkable durability, sustaining stable performance for a duration exceeding 248 h. The significant improvement in durability provides highly valuable insights for the practical application of metal-nitrogen-carbon electrocatalysts.

Nitrogen coordinated single metal site electrocatalysts (Fe-N-C) are promising alternatives to Pt-based electrocatalysts in the proton exchange membrane fuel cell (PEMFC) cathode, due to the elimination of precious metal and their remarkable catalytic activity[1–3]. However, the practical application of Fe-N-C is still unrealistic due to the fundamental constraint in electrocatalyst stability[4]. Specifically, Fe-N-C electrocatalysts usually suffer severely from demetallation of Fe from the chelation center, thus causes fast and irreversible deactivation due to structural deformation[5–7]. A typical demetallation process usually initiated by the protonation of two N atoms in the FeN₄ moiety, resulting in the formation of two N-H bonds. This process is followed by migration of the central Fe ion, together with adsorbed O₂, from the N₄ coordination site to an inactive N₂ coordination site. Finally, the Fe atom and O₂ both dissociate from the electrocatalyst to cause irreversible deactivation. This demetallation process, as reported by Wu et al., is in close relationship with site configuration. While the Fe atoms in high-spin D1 state are revealed as most active configurations, they suffer rapid deactivation during

[1]State Key Laboratory of Electroanalytic Chemistry, Jilin Province Key Laboratory of Low Carbon Chemistry Power, Changchun Institute of Applied Chemistry, Chinese Academy of Sciences, Changchun 130022, China. [2]School of Applied Chemistry and Engineering, University of Science and Technology of China, Hefei 230026, China. [3]Commercial Vehicle Development Institute, FAW Jiefang Automotive CO.LTD., Changchun 130011, China. [4]Shanghai Synchrotron Radiation Facility, Shanghai Advanced Research Institute, Chinese Academy of Sciences, Shanghai 201800, China. [5]State Key Laboratory of Rare Earth Resource Utilization, Changchun Institute of Applied Chemistry, Chinese Academy of Sciences, Changchun 130022, China. [6]National Synchrotron Radiation Laboratory (NSRL), University of Science and Technology of China, Hefei 230026, China. [7]These authors contributed equally: Jingsen Bai, Tuo Zhao, Mingjun Xu, Bingbao Mei. ✉e-mail: ywang_2012@ciac.ac.cn; zjin@ciac.ac.cn; gejunjie@ustc.edu.cn; xingwei@ciac.ac.cn

practical operation[8,9]. On the contrary, the D2 configuration, in low- or medium-spin state, convey low-activity but high-durability, due to its higher activation energy of nitrogen protonation[8,9].

Apart from the direct Fe-N chelation structure, the next nearest neighbor carbon atoms are also important in determining the electrocatalyst stability. While the Fe-N-C are thermally derived from microporous material under high temperature pyrolysis, the generation of unstable amorphous and defective carbon is unavoidable. These defective sites induce a 2-electron pathway for the oxygen reduction reaction[10-13], thereby leading to the generation of a large quantity of free radicals that subsequently attack the electrocatalyst, causing rapid electrocatalyst degradation[14-16]. While improved durability can be achieved via introducing radical scavengers, as proved by Hu et al. and in our recent work (Fe, Ce-N-C)[17,18], addressing the problem at its root is more desirable, i.e., reducing the 2-electron pathway via structure modulation of the carbon species. To this end, it is clear that developing a highly stable Fe-N-C electrocatalysts requires not only modulation in Fe coordination local structure, but also enhancement in the graphitization degree of the carbon carrier, which is of paramount importance for real world application.

In this work, we present a neoteric electrocatalyst synthesized via chemical vapor modification (CVM) method. The modification process drives the transformation in the Fe coordination structure, thereby fundamentally enhancing the durability of the electrocatalyst. Additionally, during the CVM process, the removal of amorphous carbon and defective carbon, as well as the repair of defect structures, takes place, leading to an improved local coordination environment of FeN4. This electrocatalyst exhibits exceptional stability, which has not been reported previously. Even after 200,000 cycles of accelerated aging tests, its activity has hardly diminished. The long-term steady-state test have shown that the electrocatalyst can operate reliably for over 248 h in H$_2$-air fuel cells, demonstrating excellent operational durability under working conditions. Both experimental and theoretical calculations confirm that the Fe atom transforms from the initial FeN$_4$ configuration, where four identical N atoms coordinate with Fe, to a neoteric configuration with two pyridinic and two pyrrolic N atoms (FeN$_2$+N'$_2$), which is a neoteric type of active site different from that previously reported. The FeN$_2$+N'$_2$ site possess great corrosion resistant, which could keep the activity of the electrocatalyst well retained after the durability tests.

## Results

### Electrocatalyst synthesis and physical characterization

We conduct CVM by placing the Fe-N-C electrocatalyst (synthesized via pyrolysis of FeZn-ZIF-8) to the downstream of tube furnace, which is chemical modified by the N/C containing vapors produced from cyanamide decomposition in the upstream (Fig. 1a). The final sample is further denoted as Fe-N-C$_{CVM}$. Unambiguous morphology alternation is observed upon CVM, with the edges of the dodecahedron became sharper and resembles those of crystalline ZIF, when compared with the Fe-N-C precursor (Fig. 1b, c and Supplementary Fig. 1). Moreover, the adhesion between dodecahedron particles is greatly alleviated after CVM, implying that the modification process involves both C/N source deposition on the electrocatalyst and etching/removal of unstable species to achieve the final Fe-N-C$_{CVM}$. Transmission electron microscopy (TEM) images (Fig. 1d, e and Supplementary Fig. 2) further corroborate the microstructural changes of the particles before and after CVM. Notably, while the pristine Fe-N-C electrocatalyst show obvious aggregation with the carbon boundary of the prismatic dodecahedron blurred, the Fe-N-C$_{CVM}$ electrocatalyst exhibit distinct edges and corners. As shown in Supplementary Fig. 3a, at higher relative pressures, the occurrence of capillary condensation causes the desorption isotherm to lag above the adsorption isotherm, creating a hysteresis loop. This indicates the presence of mesoporous structures in both electrocatalysts.

Interestingly, the hysteresis loop width of Fe-N-C$_{CVM}$ is greater than that of Fe-N-C, suggesting that Fe-N-C$_{CVM}$ has a more abundant mesoporous structure. The pore size distribution chart further validates our speculation. While no significant change in the micropore distribution below 2 nm in Fe-N-C$_{CVM}$ is observed in comparison to Fe-N-C, the mesopore (2–4 nm) content significantly increases (Supplementary Fig. 3b) in the former. The variation in pore structure leads to an overall decrease in specific surface area from the original 1246 m$^2$ g$^{-1}$ to 1196 m$^2$ g$^{-1}$ after CVM, corroborating the etching of unstable microporous carbon species to create mesopores. Raman spectra revealed that Fe-N-C$_{CVM}$ electrocatalyst contains a reduced proportion of defective carbon, with I$_D$/I$_G$ band intensity ratio (Fig. 1g, Supplementary Fig. 4) decrease from 1.86 to 1.50, implying a more graphitized and stable carbon structure[19,20]. The higher graphitization degree of Fe-N-C$_{CVM}$ is further revealed by X-ray powder diffraction spectra (XRD), with diffraction intensity at 25° (0 0 2 plane) significantly increased in comparison to that of Fe-N-C[19,21]. It is worth noting that no peaks related to metallic crystallographic structures are detected in XRD patterns (Fig. 1g), indicating the absence of Fe nanoparticles and clusters after CVM[22-24].

We then restored to aberration-corrected high angle annular dark-field scanning transmission electron microscopy (HAADF-STEM) (Fig. 2a, b) to further unveil the dispersion status of Fe in the sample. Single Fe sites (bright dots) uniformly dispersed in the carbon phase with randomly oriented graphitic domains, with no Fe-containing particles observed in either electrocatalysts[25]. Interestingly, an increased Fe single-atom density (Supplementary 5a–h) is noticed upon CVM, consistent with ICP results, where the Fe content increased from 0.63% in Fe-N-C to relative to 1.13% in Fe-N-C$_{CVM}$ (Supplementary Table 1). Owning to the fact that no additional Fe source is added, and that the final weight of Fe-N-C$_{CVM}$ is 40% smaller than that of the Fe-N-C precursor, we attribute the increase in Fe site density to the CVM induced etching and removal of the defective amorphous carbon. X-ray absorption spectroscopy (XAS) technique has been employed to confirm the atomic dispersion of Fe sites and determine the local coordination number. Fe K-edges in X-ray absorption near structure (XANES) spectra show that the Fe species in both Fe-N-C$_{CVM}$ and Fe-N-C electrocatalysts possess the same oxidation state (Fig. 2c). The Fourier transform of the extended X-ray absorption fine structure (FT-EXAFS) spectra of Fe atoms in both electrocatalysts existed as mononuclear centers without the presence of Fe-derived crystalline structures due to the absence of an Fe-Fe scattering path (Fig. 2d)[26,27]. However, the scattering path attributed to single Fe sites coordinated with low atomic number elements (for example, N, C and O) is significantly different, indicating a major difference in local coordination environment of the Fe species before and after CVM. Significantly, the dominant scattering path for Fe-N-C occurred at approximately 1.53 Å, while Fe-N-C$_{CVM}$ showed a dominant scattering path at 1.43 Å, indicating prominent shrinking in Fe-N bond length, probably owing to a variation in chelation environment. We then carried out quantitative least-squares EXAFS fitting analysis to obtain the local chelation parameter (Supplementary Fig. 6 and Supplementary Tables 2, 3). Interestingly, the best fitting results of Fe-N-C$_{CVM}$ contain two distinct scattering paths, each with a coordination number of 2 (Fig. 2e and Supplementary Fig. 7), in sharp contrast with a single path best fitting (Fig. 2e, Supplementary Fig. 8) of the Fe-N-C electrocatalyst. It is well known that N coordinated with Fe includes pyridinic N and pyrrolic N. The results provide initial evidence supporting the transformation of the active site configuration in the electrocatalyst from the original FeN$_4$ to the FeN$_2$+N'$_2$ following CVM treatment. The EXAFS wavelet transforms (WT) plot is another powerful tool for distinguishing the coordination environment, which suggests that Fe-N of Fe-N-C$_{CVM}$ shows up in a higher wavenumber k range than Fe-N-C. Atomic scattering path exhibited two different radial distance of

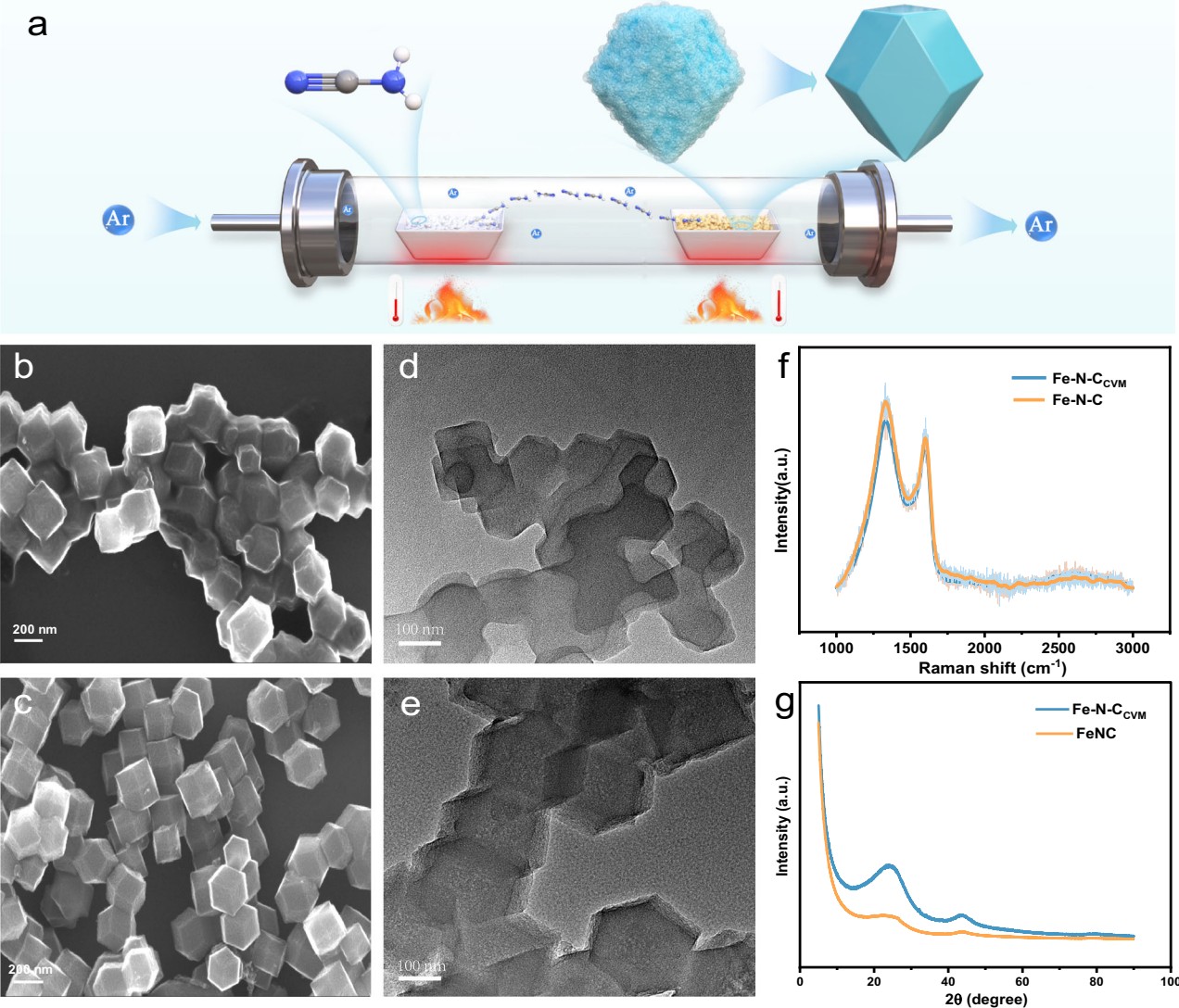

**Fig. 1 | Synthesis and characterization of Fe-N-C and Fe-N-C$_{CVM}$. a** Schematic depiction of the synthesis of Fe-N-C$_{CVM}$ electrocatalyst. Scanning electron microscopy (SEM) images of the Fe-N-C (**b**) and Fe-N-C$_{CVM}$ (**c**) electrocatalysts. TEM images of the Fe-N-C (**d**) and Fe-N-C$_{CVM}$ (**e**) electrocatalysts. **f** Raman spectra of the Fe-N-C and Fe-N-C$_{CVM}$ electrocatalysts. **g** XRD patterns of the Fe-N-C and Fe-N-C$_{CVM}$ electrocatalysts.

Fe-N-C$_{CVM}$ and Fe-N-C, again indicating that the local coordination environment has been changed after CVM (Fig. 2g, h, i)[28]. Moreover, the type and content of N atoms also contain the information about the type of active site. We therefore employed N XANES spectra (Fig. 2f) and X-ray photoelectron spectroscopic (XPS) N$_{1s}$ spectrum (Supplementary Fig. 9) to characterize the type and content of N atoms. In N K-edge XANES spectra, there are two major N 1s → π* at 399.5 eV (d$_1$) and 401 eV (d$_2$) corresponding to pyridinic-N and pyrrolic-N (N bonded to two carbon atoms, C-N-C), and one major N 1 s → σ* at 408 eV (d$_3$) corresponding to graphitic N (N bonded to three carbon atoms, N-C$_3$)[29–33]. As shown in Fig. 2f, the Fe-N-C$_{CVM}$ exhibits a much higher d$_1$ peak intensity and a lower d$_2$ intensity, indicating the variation in Fe coordination environment after CVM. High-resolution N 1 s spectra further corroborate the variation in N structure, with peak intensity associated with pyridinic N (~398.5 eV) increased in the Fe-N-C$_{CVM}$[19,34,35]. Such alternation is favorable for the transformation in active site configuration, i.e., from pyridinic N is highly unlikely, as a large proportion of pyrrolic N still exist in the sample, and it is statistically unrealistic to get a full conversion for all four nitrogen atoms into the pyridinic structure. Combining all together, we infer that the alternation in Fe scattering path might be associated with the change in Fe coordinated N structure, i.e., the

partial transformation from pyrrolic N to pyridinic N and the formation of a FeN$_2$+N'$_2$ neoteric structure.

## Electrochemical measurement

Subsequently, we evaluated the electrocatalytic oxygen reduction reaction (ORR) activity of Fe-N-C and Fe-N-C$_{CVM}$ using a rotating ring disk electrode (RDE) in 0.1 M HClO$_4$. The N$_2$ cyclic voltammetry of Fe-N-C and Fe-N-C$_{CVM}$ show similar double-layer capacitance, of both electrocatalysts did not significantly differ before and after CVM, indicating similar electrochemical active surface before and after CVM process (Fig. 3a). Meanwhile, the oxygen reduction on the two electrocatalysts also show similar half-wave potential (E$_{1/2}$) between Fe-N-C and Fe-N-C$_{CVM}$ (Fig. 3b), indicating similar activity of the two electrocatalysts. The stability of the two electrocatalysts, however, are completely different, with Fe-N-C$_{CVM}$ exhibits much higher stability than the pristine Fe-N-C. Notably, 200,000 cycles accelerated stress tests (ASTs) induce no performance decay in Fe-N-C$_{CVM}$ (Fig. 3c), which is even better than the state-of-the-art commercial Pt/C catalyst (Johnson Matthey Company) in terms of cycling stability (Supplementary Figs. 10 and 11). To the contrary, the Fe-N-C suffers continual loss in E$_{1/2}$ by 24, 37, and 44 mV, after ASTs conducted for 10,000 cycles, 30,000 cycles, and 50,000 cycles, respectively (Fig. 3d), in line with literature (Supplementary Table 4). In

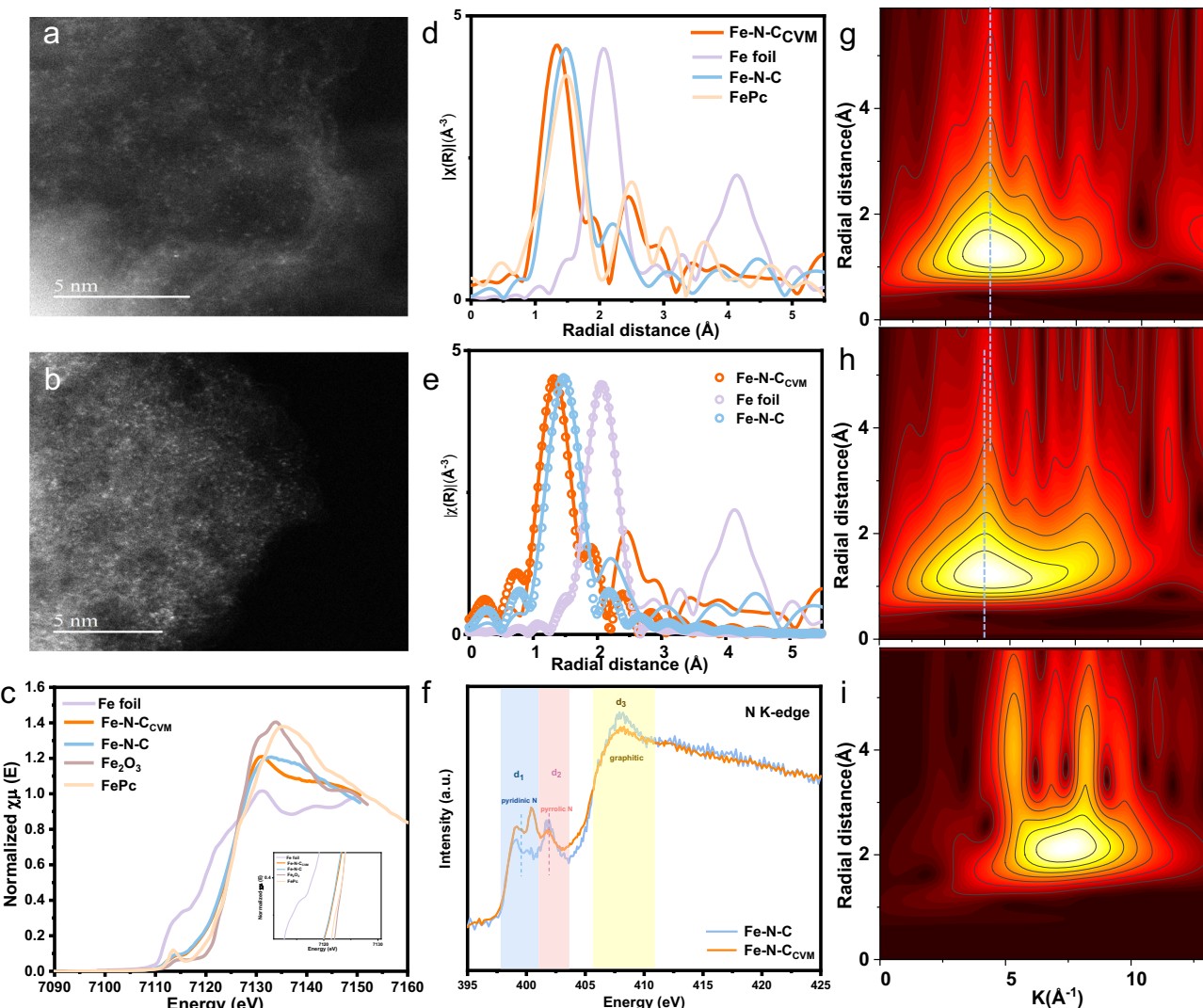

**Fig. 2 | The configuration of active sites in the electrocatalysts. a** HAADF-STEM images of the Fe-N-C. **b** HAADF-STEM images of the Fe-N-C$_{CVM}$. **c** Normalized XANES spectra of Fe K-edge for Fe foil, Fe-N-C$_{CVM}$, Fe-N-C, Fe$_2$O$_3$ and FePc. **d** The $k^3$-weighted EXAFS spectra for Fe foil, Fe-N-C$_{CVM}$, Fe-N-C and FePc. **e** EXAFS fitting results of Fe foil, Fe-N-C$_{CVM}$, Fe-N-C. **f** N K-edge XANES spectra of Fe-N-C$_{CVM}$ and Fe-N-C. Wavelet transform of Fe K-edge EXAFS data for (**g**) Fe-N-C$_{CVM}$, (**h**) Fe-N-C and (**i**) Fe foil.

terms of mass activity, Fe-N-C$_{CVM}$ presents a higher value of 10 A g$_{electrocatalyst}$$^{-1}$ than Fe-N-C (4.17 A g$_{electrocatalyst}$$^{-1}$). Additionally, after CVM, the Tafel slope of the electrocatalyst has changed from the original 88 mV dec$^{-1}$ to 58 mV dec$^{-1}$ (Supplementary Fig. 12). This implies that compared to Fe-N-C, Fe-N-C$_{CVM}$ exhibits a faster kinetic reaction rate. Moreover, we observed lower hydrogen peroxide yield and higher electron transfers number of Fe-N-C$_{CVM}$ via rotating ring-disk electrode (RRDE) measurement, which may partially contribute to the high stability of the electrocatalyst (Fig. 3e, f). We attribute the excellent durability as well as the lowered H$_2$O$_2$ yield to the transformation of active site structure from FeN$_4$ to FeN$_2$+N'$_2$ after the CVM process. Specifically, the N group of Fe-N-C electrocatalyst may be more readily protonated with H to form N-H bonds, due to not undergoing CVM, which will lead to weakened Fe-N interaction and easy demetallation of the active sites. Meanwhile, the exposure of Fe-N-C to H$_2$O$_2$ leads to significant decrease in turnover frequency (TOF) from 0.67 to 0.21 e site$^{-1}$ s$^{-1}$ (0.8 V at pH = 1) (Supplementary Figs. 13 and 14), ascribable to carbon surface oxidation and thereby weakened O$_2$-binding on iron-based sites. On the contrary, the Fe-N-C$_{CVM}$ sustains its high TOF value of -1.0 e site$^{-1}$ s$^{-1}$ (0.8 V at pH = 1) after H$_2$O$_2$ exposure, presumably owning to the overcome of the demetallation problem caused by N protonation (Supplementary Fig. 15

and Supplementary Fig. 16)[36–38]. Energy-dispersive X-ray spectroscopy (EDS) analysis shows that the oxygen content of the Fe-N-C electrocatalyst increased significantly from 2.09% at the beginning to 6.39% after durability testing. However, the oxygen content of the Fe-N-C$_{CVM}$ electrocatalyst exhibits an opposite trend, with O even decreased from 2.08% to 1.74% after durability test (Supplementary Table 5). We further conducted in situ differential electrochemical mass spectroscopy (DEMS) to compare the initial oxidation potentials of Fe-N-C and Fe-N-C$_{CVM}$. As shown in Supplementary Fig. 17, the initial oxidation potential of Fe-N-C is 0.71 V, significantly lower than that of Fe-N-C$_{CVM}$ (0.86 V). This could be attributed to the higher graphitization degree in the carbon substrate of Fe-N-C$_{CVM}$ achieved through CVM process, which contributes to the higher stability of the electrocatalyst.

We then investigated the electrocatalyst structure after the accelerated stress tests to probe into the essential difference between the two electrocatalysts. The Fe-N-C sample showed obvious agglomeration, with the edges of the dodecahedron became more blurred, indicative of higher degree of corrosion (Supplementary Fig. 18a). The Fe-N-C$_{CVM}$, on the contrary, well retained its dodecahedron structure resembles those of the electrocatalysts before durability testing (Supplementary Fig. 18b). TEM images further corroborate the intact

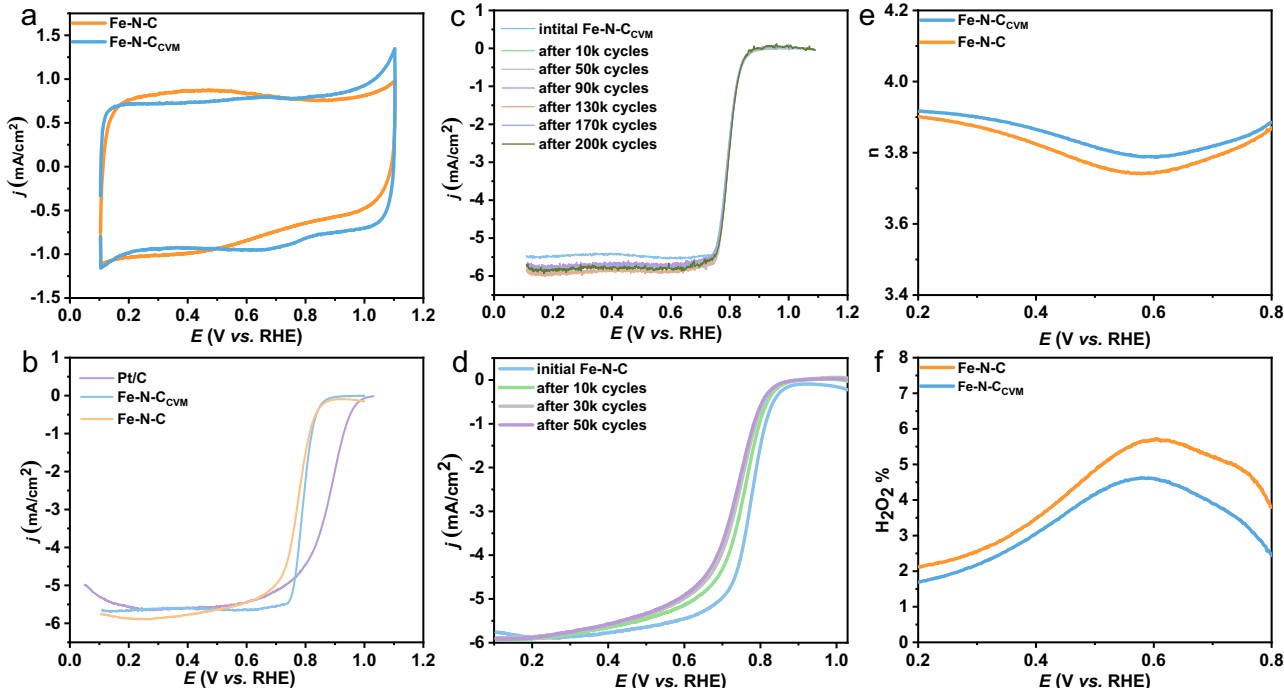

**Fig. 3 | Electrocatalytic performance of Fe-N-C and Fe-N-C$_{CVM}$ in three-electrode setup. a** Cyclic voltammetry curves of Fe-N-C and Fe-N-C$_{CVM}$ in N$_2$ saturated 0.1 M HClO$_4$ with a sweep rate at 10 mV/s. **b** ORR polarization curves of Fe-N-C and Fe-N-C$_{CVM}$ in O$_2$ saturated 0.1 M HClO$_4$ with a sweep rate at 10 mV/s. ORR polarization curves. (**c**) Fe-N-C$_{CVM}$ and (**d**) Fe-N-C before and after different potential cycles between 0.6 and 1.0 V in O$_2$ saturated 0.1 M HClO$_4$ with a sweep rate at 10 mV/s. **e** electron transfer number and (**f**) H$_2$O$_2$ yield of Fe-N-C and Fe-N-C$_{CVM}$.

structure of Fe-N-C$_{CVM}$ after tests, while the outermost catalytic layer of Fe-N-C underwent severe structure evolution to form amorphous carbon state (Supplementary Fig. 18c, d).

With the high electrochemical performance achieved, we next turned to probe the activity and stability of the Fe-N-C$_{CVM}$ in practical PEMFC measurements. Promisingly, Fe-N-C$_{CVM}$ cathode generated 130 mA cm$^{-2}$ at 0.8 V$_{iR-free}$ (Fig. 4a)[9] and 26 mA cm$^{-2}$ at 0.9 V$_{iR-free}$ under H$_2$-O$_2$ mode. The peak power density reached 450 mW cm$^{-2}$ under H$_2$ air conditions (Fig. 4b), on par with the best PGM-free electrocatalysts (Supplementary Table 6) recently reported[39]. To the contrary, the Fe-N-C cathode conveys a much lower performance of 36 mA cm$^{-2}$ at 0.8 V, accompanied with a power density of merely 249 mW cm$^{-2}$. Specifically, in terms of durability, Fe-N-C$_{CVM}$ demonstrated excellent performance, with only a 4.5% decrease in power density after 20 k AST cycles. After 30k cycles, the voltage at 0.8 A cm$^{-2}$ only decreased by 20 mV (from 0.486 V to 0.466 V vs RHE), successfully meeting the DOE 2025 target (≤30 mV loss at 0.8 A cm$^{-2}$). More excitingly, 80% of the original power density was retained after 70k AST cycles (Fig. 4c) with cell voltage only decrease by 48 mV, indicating robust durability. In contrast, Fe-N-C experienced 40% decrease in power density after 70 k accelerated aging test cycles with cell voltage decreased by 171 mV (Fig. 4d). Such distinctive variation in cell durability further corroborates the much enhanced stability of the electrocatalysts. More excitingly, the constant voltage tests (0.67 V and 0.55 V in H$_2$/air) demonstrate the promising stability feature of Fe-N-C$_{CVM}$ against Fe-N-C. Specifically, a rigorous aging test of 100 hours at 0.67 V for Fe-N-C$_{CVM}$ only leads a current density attenuation by 10%, and 248 h of testing leads to retained performance by 63%, which is among the best stability achieved on Fe based electrocatalysts (Supplementary Table 7). The constant voltage tests at 0.55 V also further demonstrate the high stability of the electrocatalysts, with more than 210 h of continuous operation further suggest the excellent stability. The Fe-N-C cathode, however, suffered a significant and rapid performance loss during the stability test of a constant voltage at 0.67 V, losing 71% of its

current density at 0.6 V after 43 h under H$_2$ air (Fig. 4e). Owing to the great durability of neoteric active sites, the Fe-N-C$_{CVM}$ electrocatalyst showed excellent performance under harsh MEA test conditions.

To explore the configuration of active sites in the electrocatalyst (Fe-N-C$_{CVM}$-s) after ASTs, XAS technique has been employed once again (Supplementary Fig. 19). The FT-EXAFS spectra show that the dominant scattering path for Fe-N-C$_{CVM}$ and Fe-N-C$_{CVM}$-s both occurred at approximately 1.43 Å, indicating that the Fe-N bonds in the electrocatalyst did not undergo significant changes after ASTs (Supplementary Fig. 20). Interestingly, the optimal fitting results of EXAFS for Fe-N-C$_{CVM}$-s include two distinct scattering paths, each with a coordination number of 2, consistent with Fe-N-C$_{CVM}$, indicating that the active site configuration of Fe-N-C$_{CVM}$ did not change after ASTs (Supplementary Fig. 21 and Supplementary Table 8). Moreover, as shown in XPS N$_{1s}$ spectrum, there is no significant change in the content of pyridinic nitrogen and pyrrolic nitrogen in Fe-N-C$_{CVM}$-s compared to Fe-N-C$_{CVM}$, once again confirming that the active site configuration of the electrocatalyst has not changed (Supplementary Figs. 9 and 22).

### Mechanisms for enhancing the stability of neoteric active site

We then resorted to density functional theory (DFT) calculations to unveil the structural reasons for the high durability of the different active sites. According to the FT-EXAFS and N K-edge XANES spectra, variation in active site configuration from FeN$_4$ to FeN$_2$+N'$_2$ might be the reason for the enhanced stability due to CVM. To this end, we built up different models with Fe coordinated 2 pyrrolic N and 2 pyridinic N (Supplementary Figs. 23 and 24). The structural stability was firstly probed, with formation energy ($E_f$) of different active site configurations being calculated, in order to pick up the most thermodynamically stable structure. Among the configurations, $E_f$ of FeN$_2$+N'$_2$ with the mix configuration shows a significantly more negative $E_f$ (−1.651 V) compared to other structures, indicating its thermodynamic favorability and stability (Fig. 5a and Supplementary Table 9). Besides, a shorter Fe-

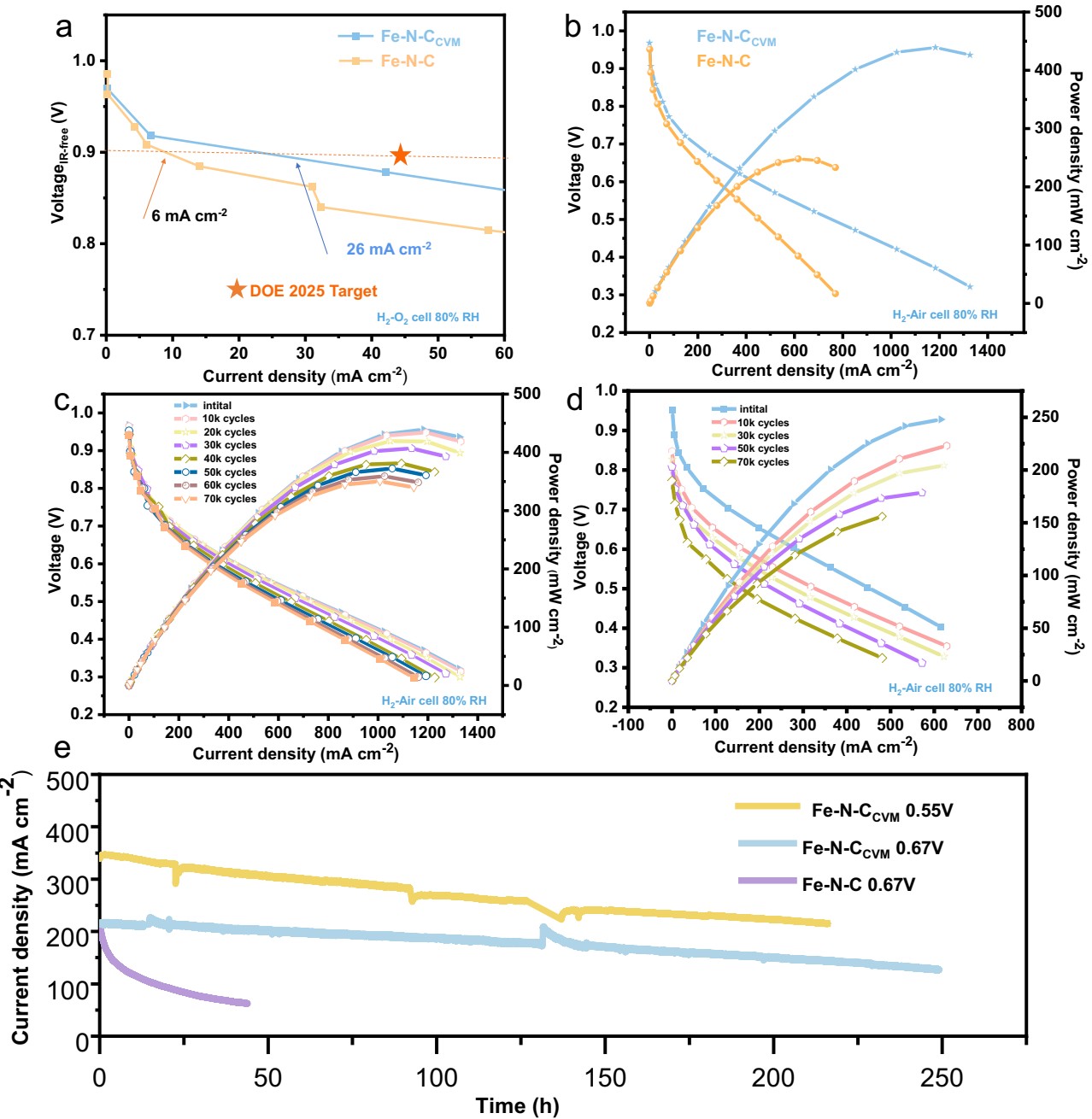

**Fig. 4 | MEA performance of Fe-N-C and Fe-N-C_CVM cathode electrocatalyst.**
**a** Determination of Fe-N-C_CVM and Fe-N-C at 0.9V_iR-free under 1 bar H_2-O_2.
**b** Polarization and power density curves of Fe-N-C_CVM and Fe-N-C. **c** Polarization and power density curves of Fe-N-C_CVM before and after different potential cycles. **d** Polarization and power density curves of Fe-N-C before and after different potential cycles. **e** Long-term fuel cell tests for Fe-N-C_CVM and Fe-N-C electrocatalysts under H_2-Air conditions at a constant potential of 0.67 V. Test conditions: cathode loading 4.0 mg cm$^{-2}$ for Fe- N-C_CVM and Fe-N-C, anode loading 0.1mg_Pt cm$^{-2}$, Nafion 212 membrane, 80 °C, 80 relative humidity (RH) and 1.0 bar H_2-Air at flow rates of 300 ml min$^{-1}$.

N bond length implies higher bond energy, which can make the active center Fe ion less prone to dissociation, leading to enhanced durability. DFT calculations revealed that the Fe-N bond length in the mix configuration (1.97 Å) is significantly shorter than that in the S1 configuration (Fe-N_4 site coordinated with four pyrrolic nitrogen atoms, 2.09 Å), which is highly consistent with the results obtained from EXAFS analysis.

To validate the intrinsic activity of the two active sites, the Gibbs free energy for ORR at $U_{RHE} = 1.23$ V was calculated (Fig. 5b, Supplementary Figs. 25 and 26). The electrochemical step associated with OH* adsorption was identified as the rate-determining step for both

electrocatalysts. It was observed that the overpotential of FeN_2+N'_2 in Fe-N-C_CVM (0.88 eV) is lower than FeN_4 in Fe-N-C (1.05 eV), indicating a weakened OH* binding and enhanced the reaction kinetics in Fe-N-C_CVM. These results suggest that the neoteric active site can optimize the adsorption of oxygen species and further enhance the ORR activity. The dissociation of metal centers exerts a profound impact on the performance of electrocatalysts. To further explore reasons for the improved durability of electrocatalyst, we examined the demetallation energy of the electrocatalyst. Under acidic conditions, the demetallation process of the electrocatalyst often arises from the protonation of nitrogen (N) atoms, resulting in the formation of two N-H bonds.

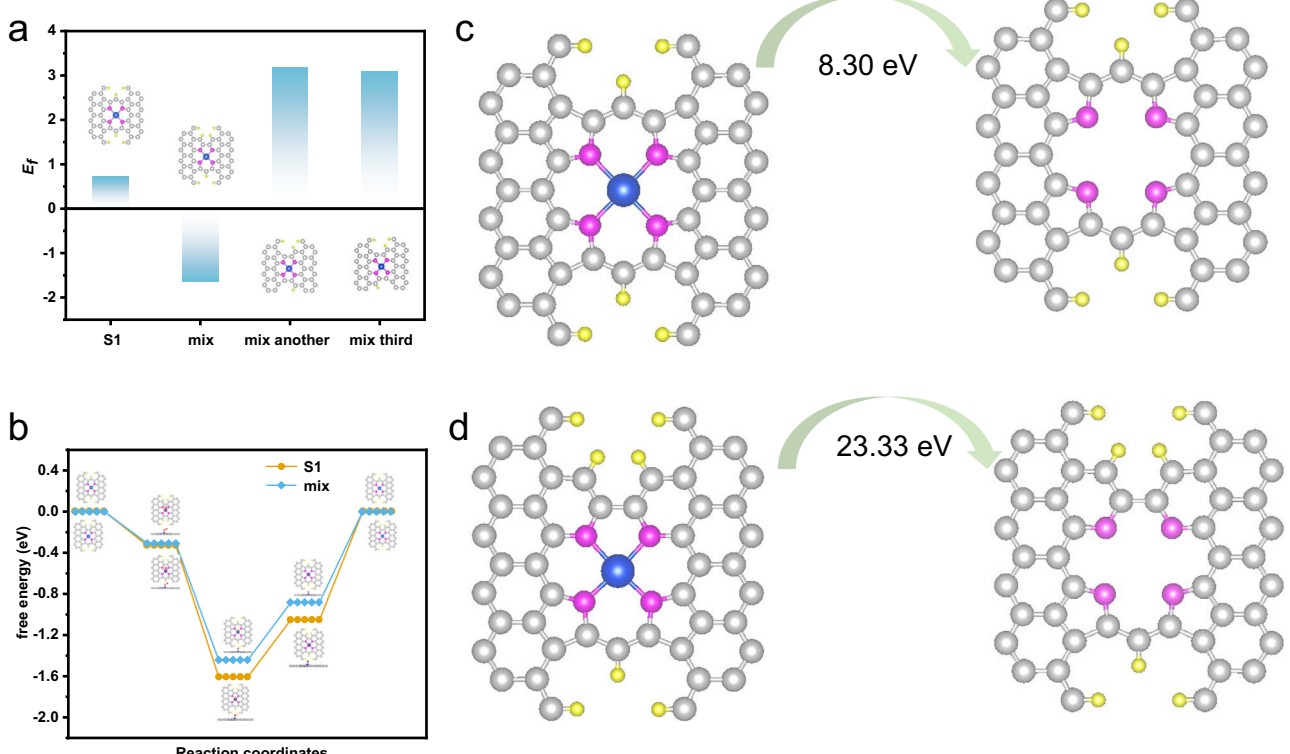

**Fig. 5 | DFT calculations to elucidate the activity and durability of FeN$_4$ and FeN$_2$+N$_2$'. a** The formation energy of various active sites' configuration. **b** Gibbs free energy diagrams at 1.23 V on Fe FeN$_4$ and FeN$_2$+N'$_2$. Illustration of (**c**) S1-type FeN$_4$ site evolve into N$_4$C$_{12}$ and (**d**) FeN$_2$+N'$_2$ evolve into N$_4$C$_{11}$.

Subsequently, the Fe ion and absorbed O$_2$ occur from N$_4$ coordination into N$_2$ coordination and eventually Fe ion with absorbed O$_2$ dissociate from the electrocatalyst. The results indicate that the metal leaching process of the mix configuration exhibits a much more positive Gibbs free energy (23.33 eV) than FeN$_4$ (8.30 eV), suggesting the huge enhancement in structure stability (Fig. 5c, d).

## Discussion

In summary, we have developed an electrocatalyst with neoteric active site configuration through CVM. During the process of CVM, the active site transformed from the traditional FeN$_4$ configuration to the FeN$_2$+N'$_2$ configuration, resulting in a substantial improvement in durability. Excitingly, the Fe-NC$_{CVM}$ electrocatalyst exhibits non-observable attenuation in half-wave potential after undergoing 200,000 cycles of AST tests. Moreover, PEMFC tests reveal high current density of 26 mA cm$^{-2}$ at 0.9 V$_{iR-free}$ for Fe-N-C$_{CVM}$, with a power density of up to 450 mW cm$^{-2}$ achieved under H$_2$-air mode with 1 bar back pressure. Even further, the cell can operate sustainably for over 248 h under demanding conditions at 0.67 V, demonstrating its immense potential for practical applications. This study significantly enhances the durability of the electrocatalyst, providing valuable insights for the development of highly stable non-precious metal electrocatalysts in the future.

## Methods
### Materials
Zinc nitrate hexahydrate (Zn (NO$_3$)$_2$·6H$_2$O, ≥99.99%), 2-methylimidazole (98%), Iron (III) acetylacetonate, Melamine (Fe(acac)$_3$, 99%) were obtained from Aladdin company. Commercial state-of-the-art 20 wt% Pt/C (Johnson Matthey Company, HiSPEC™ 3000) was used as the benchmark for comparison. 5 wt% Nafion ionomer, Nafion 212 membrane were purchased from DuPont Co. Methanol (CH$_3$OH, 99.5%), Ethanol (C$_2$H$_5$OH, 99.5%), Hydrochloric acid (HCl, 37%) were provided by Beijing Chemical Works. Ultrapure water (Millipore, 18.25 MΩ cm) was used throughout all experiments.

### Synthesis of FeZn-ZIFs
3.0 g Zn(NO$_3$)$_2$·6H$_2$O and 440 mg Iron(III) acetylacetonate were dissolved in 40 mL methanol and then mixed with 80 mL methanolic solution of 6.5 g 2-methylimidazole by stirring for 24 h at room temperature. The as-obtained reddish-brown precipitate was centrifuged and washed with ethanol three times and then dried in a vacuum oven at 55 °C for 12 h.

### Synthesis of Fe-N-C
The FeZn-ZIFs were subsequently treated at 950 °C in a tube furnace with a heating rate of 5 °C/min under hydrogen argon mixture (H$_2$/Ar is 10/90) flow for 2 h. After cooling down to room temperature, the carbon material was etched by 1 M HCl for 12 h. Then the products were washed with ultrapure water and dried thoroughly at 55 °C to obtain Fe-N-C electrocatalyst.

### Synthesis of Fe-N-C$_{CVM}$
2.0 g cyanamide was placed in a boat in a tube upstream of the Ar flow; 60 mg Fe-N-C electrocatalyst was placed in a boat in a tube downstream of the Ar flow. The Fe-N-C electrocatalyst was heated to 900 °C for 2 h with a heating rate of 5 °C/min and the cyanamide was rapidly heated to 700 °C with a heating rate of 10 °C/min. After cooling down to room temperature, Fe-N-C$_{CVM}$ single-atom electrocatalyst can be obtained.

### Physical characterizations
Scanning electron microscopy (SEM) measurements were performed with an XL 30 ESEMFEG field emission scanning electron microscope. TEM, high resolution transmission electron microscopy, high-annular darkfield scanning transmission electron microscopy (STEM) and element mapping analysis were conducted on a Philips TECNAI G2 electron microscope operating at 200 kV. X-ray photoelectron spectroscopy (XPS) measurements were performed with Mg Kα radiation source on KratosXSAM-800 spectrometer. X-ray diffraction (XRD) measurements

were performed with a PW-1700 diffractometer using a Cu Kα ($\lambda = 1.5405$ Å) radiation source (Philips Co.) The HAADF-STEM images were imaged by using a Titan 80-300 scanning/transmission electron microscope operated at 300 kV, equipped with a probe S3 spherical aberration corrector. X-ray absorption fine structure (XAFS) spectra was recorded on the beam line at the Shanghai Synchrotron Radiation Facility (SSRF), Shanghai Institute of Applied Physics, China. The powder electrocatalysts were characterized in transmission mode, and the energy of Fe K edge were calibrated using standard Fe foil, respectively. The XAFS raw data were background-subtracted and normalized by the ATHENA program. Least-squares curve-fitting analysis of the EXAFS χ(k) data and Fourier transforms were carried out using the ARTEMIS program.

The spectra were calibrated, averaged, pre-edge background subtracted, and post-edge normalized using the Athena program in the IFEFFIT software package. Using $E_0$ as the zero point, the pre-edge was defined within the range of $-150$ eV to $-30$ eV, with a normalization range of 150 eV to 670 eV. The plotting k-weights is 3. To prevent data distortion, spectral curves were not subjected to smoothing. The Δk range/nm$^{-1}$ for the FT into R-space of Fe-N-C and Fe-N-C$_{CVM}$ is 3–10 Å$^{-1}$. The ΔR range for the curve fitting in R-space is 1–2.2 Å.

### Electrochemical measurements

The electrochemical performance of all electrocatalysts was evaluated using the 750E Bipotentiostat (CH Instruments). When testing Fe-based electrocatalysts, a saturated calomel electrode (SCE) was employed as the reference electrode, a graphite rod as the counter electrode, and a rotating ring-disk electrode (RRDE) coated with the electrocatalyst ink as the working electrode. For testing the commercial Pt/C electrocatalyst, to prevent poisoning of Pt, a reversible hydrogen electrode was used as the reference electrode, while the other conditions remained unchanged.

Nonplatinum electrocatalyst ink: Mix 50 μL of Nafion (5 wt%) with 950 μL of ethanol, then add 5 mg of electrocatalyst and sonicate until the electrocatalyst is well dispersed. This process should be carried out at a temperature below 25 °C.

20 wt% Pt/C electrocatalyst ink: First, mix 50 μL of Nafion (5 wt%) solution with 400 μL of H$_2$O, then add 5 mg of 20 wt% commercial Pt/C, and finally add 550 μL of isopropanol. Sonicate the ink until uniform and coat it onto the RRDE, ensuring a Pt loading of 20 μg cm$^{-2}$.

The electrolyte for ORR testing is 0.1 M HClO$_4$, and the measurements are conducted at room temperature (~25 °C). RRDE measurements were conducted by liner sweep voltammetry (LSV) from 1.1 V to 0.1 V at a scan rate of 10 mV s$^{-1}$ at 1600 rpm, while the ring electrode was held at 1.3 V vs. RHE. The number of transferred electrons (n) and the yield of H$_2$O$_2$ are calculated using the following equations:

$$n = \frac{4I_D}{I_D + (I_R/N)} \quad (1)$$

$$\%H_2O_2 = 100 \frac{2I_R/N}{I_D + (I_R/N)} \quad (2)$$

where $I_D$ is the faradaic current at the disk, $I_R$ the faradaic current at the ring and N (0.37) is the H$_2$O$_2$ collection coefficient at the ring.

The kinetic current density (i) of the oxygen reduction process is calculated using the Koutecky-Levich (K-L) equation:

$$\frac{1}{i} = \frac{1}{i_k} + \frac{1}{i_l} \quad (3)$$

where i represents the measured current density, $i_k$ is the kinetic current density and $i_l$ is the limitting currenrt density.

The ORR results are obtained by subtracting the measured current in 0.1 M HClO$_4$ electrolyte saturated with N$_2$ to remove the capacitive currents. The accelerated durability tests (ADTs) were performed at room temperature (~25 °C) in O$_2$-saturated 0.1 M HClO$_4$ solution. The cyclic potential sweeps were applied to RHE from 0.6 to 1.0 V at a scan rate of 200 mV s$^{-1}$ for 200,000 cycles, and the initial and final LSV curves were collected.

### Fuel cell MEA tests

For the cathode, firstly, 320 μL Nafion solution (5 wt%) was thoroughly mixed with 470 μL of ultrapure water. Then, 25 mg of electrocatalyst and 470 μL of isopropanol were added to the mixture. The ink was sonicated for one hour at around 25 °C. Subsequently, it was spray-coated onto carbon paper with an electrocatalyst loading of 3 mg cm$^{-2}$. As for the anode, Nafion solution (5 wt%) mixed thoroughly with ultrapure water, and then 20% wt commercial Pt/C and isopropanol were added to the solution. The ink was sonicated until uniform and then spray-coated onto carbon paper, maintaining a Pt loading of 0.1 mg$_{Pt}$ cm$^{-2}$. The anode gas diffusion electrode, cathode gas diffusion electrode and a Nafion 212 membrane (DuPont Co) were hot-pressed together to prepare the MEA. The hot-pressing conditions were set at 135 °C, 1 ton pressure, and 120 s duration. The area of the Nafion 212 membrane is 3*3 cm$^2$ with a thickness of 50 μm, and it was used directly without any treatment. The MEA was measured by a fuel cell test station (Scribner 850e) with UHP-grade H$_2$ and O$_2$ or air at 80 °C, 80%RH. The flow rate for both the cathode (O$_2$, air) and anode (H$_2$) were set at 300 ml min$^{-1}$. Under constant conditions with cell voltages of 0.55 V and 0.67 V, the flow rates of air and H$_2$ were maintained at 300 ml min$^{-1}$ for 220 h and 248 h, respectively, for the durability test of the H$_2$-air fuel cell.

## Data availability

The data that support the findings of this study are available within the article and its Supplementary Information files. All other relevant data supporting the findings of this study are available from the corresponding authors upon request. Source data (SEM, TEM, XRD, BET, Raman, HAADF-STEM, XAFS, XPS, Electrochemical measurement, DFT) are provided with this paper. Source data are provided with this paper.

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

## Acknowledgements
Thank the National Key R&D Program of China (2022YFB4004100), the National Natural Science Foundation of China (22272161, 22179126), the Jilin Province Science and Technology Development Program (YDZJ202202CXJD011) for financial support. Thank the Shanghai Synchrotron Radiation Facility for conducting the X-ray absorption spectroscopy experiments at BL08U1A and BL14W1 station. Part of the computational time is supported by the High Performance and Computing Center of Jilin University, Jilin Province, Net-work and Computing Center in Changchun Institute of Applied Chemistry, Chinese Academy of Sciences.

## Author contributions
J.B. and J.G. conceived the experimental ideals. J.B. and T.Z. synthesized the electrocatalysts. J.B., Mingjun.X., B.M., Zhao.J. and J.G. performed the physical characterization and results discussion. J.B., B.M., and Y.W. conducted the theoretical calculations. L.Y., Z.S., S.Z., Zheng.J., Mei-ling.X. and C.L. help analyzed the results. J.B., Zhao.J., J.G. and W.X. wrote the manuscript. All the authors commented on the results and approved the final version of the manuscript.

## Competing interests
The authors declare no competing interests.
