## [Peer Review File · Nature Communications]

REVIEWER COMMENTS

Reviewer #1 (Remarks to the Author):

Interesting paper on FeNC electrocatalysts for ORR, tested in acid medium, and air/H₂ PEMFC. Main comments:

- First of all the words “catalyst” must be written as “electrocatalyst” to be precise.
- Figure 1b/d/c/e (and 1S_all, S10a/b): length bar missing, please add.
- Figure 2 and related comments on the nature of the active site: the authors demonstrated that the active site is FeN₂+N'₂ in the sample prepared by CVM. How many times the CMV synthesis has been repeated, providing the same F₂N₂+N'₂ results? This is an important point considering the good performance obtained with the FeNC_CVM synthesis: repeatability.
- Figure 3: add in the caption the nature of the electrolyte (0.1 M HClO₄) and the mV/s at which each test has been performed. It would be useful to add also the performance of a commercial Pt/C as a reference. The authors claim that the better stability of FeNC_CVM is due to the presence of FeN₂+N'₂ active sites. In which way exactly these sites make more stable the process? Which reaction mechanism is hypothesized here?
- To discuss about decreasing of TOF without measuring it is a nonsense. Or the authors measure is (TOF and SD, site density), of this is pure speculation. It can be avoided if the measurements cannot be performed. There are several papers that can be used as a reference. A few examples: Appl Catal B Environ 2005, 56:9–35. Nat Commun 2016, 7:1–7. Current Opinion in Electrochemistry 2021, 27:100683.
- Please provide the Fe content before/after stability tests (as done for O and N).
- Figure 4a: DOE target values: the one indicated by DOE is 44 mA/cm² @ 900 mV IR-free (and not > 60 at 0.8), as stated in the paper Solid State Ionics 319 (2018) 68–76. Please, refer to the correct one (also by citing this reference), and specify if the value is IR-free or not.

Reviewer #2 (Remarks to the Author):

In their manuscript entitled “Monosymmetric Fe-N₄ sites enabling super durable PEMFC cathode by chemical vapor modification” the authors report the high durability of a modified FeN₄-type electrocatalyst for ORR. In the chemical vapor modification (CVM) step the authors claim the transformation of the FeN₄ sites into a FeN₂+N'₂ configuration resulting in a nonsymmetric coordination environment of the atomically dispersed Fe species. While the manuscript provides sufficient data for the modification process of the original substrate by CVM and the newly prepared catalyst shows high durability according to the authors data, publication in Nature Communications cannot be recommended due to the following reasons.

- The only noteworthy result is the high catalyst durability
- The synthesis method isn't entirely new, CVD is an established method for nanoparticle and surface modification (Zhao, X., Wei, C., Gai, Z. et al. Chemical vapor deposition and its application in surface modification of nanoparticles. Chem. Pap. 74, 767–778 (2020). <https://doi.org/10.1007/s11696-019-00963-y>)
- The catalyst ORR performance under PEMFC measurements is reportedly 360 mW cm⁻² under H₂/air

conditions, which to the opposite of the authors claim is not on par with the best PGM-free catalysts, comparing to the values given in Supplementary Table 6 of the manuscript. Fe-AC-AC-CVD reaches 560 mW cm⁻², TPI@Z8(SiO₂)-650-C reaches 420 mW cm⁻².

- To be published in Nature Communications I would expect the authors reporting a significant higher power density as for example 700-800 mW cm⁻².
- Jiao et al. reported a Fe-N-C ORR activity of 33 mA cm⁻² at 0.90 V (iR-corrected; i, current; R, resistance) in a H₂-O₂ proton exchange membrane fuel cell at 1.0 bar and 80 °C. (Jiao, L., Li, J., Richard, L.L. et al. Chemical vapor deposition of Fe-N-C oxygen reduction catalysts with full utilization of dense Fe-N₄ sites. Nat. Mater. 20, 1385–1391 (2021). <https://doi.org/10.1038/s41563-021-01030-2>). The authors should provide similar data for their catalyst under identical conditions, especially the catalyst activity at 0.90 V.
- Most importantly, two of the standard parameters for ORR catalyst activity, the mass activity in A gcatalyst⁻¹ and Tafel slope in mV dec⁻¹ measured by RDE is missing. This value is essential for catalyst evaluation. For example, Menga et al., report a mass activity of 2.7 ± 0.3 A g⁻¹ and Tafel slope of ≈ 60 mV dec⁻¹ for their Fe-N₄ catalyst containing tetrapyrrolic FeN₄ sites. (Angew. Chem.Int. Ed.2022,61, e20220)
- Another important parameter for highly porous carbon materials used as electrocatalysts, the apparent surface area as for example measured by BET is missing. A standard carbon support material like Ketjen black™ for example is reported to have 300 or 600 m² g⁻¹. How does the authors catalyst compare to this value or other Fe-N-C based catalysts?
- Some of the figures require revision, as for example figures 4 a and b, where current densities are listed in A cm⁻² and W cm⁻². Figures 3 c and d “1w cycles”, “3w cycles”, and “5w cycles”.

Other comments to the authors

L23: “increased graphitization degree enhancement” What is enhanced and what increased

L46: “high-spin D1 state” Does this refer to 57Fe Mössbauer spectroscopy as reported by Wang et al.? (Angew. Chem.Int. Ed.2023,62, e2023042)

L97: “mesopores” From SI Figure 3 it appears that 2 different pore sizes exist in Fe-N-CCVM. The authors should specify and discuss this.

L104: “Figure 1d” should be 1g to my understanding

L116: “Supplementary 5a-h” The Fe distribution is hardly observable

L128: “Figure 2d” Same colors should be used for same species, very confusing

L139: “shrinking in Fe-N bond length” Does this reflect a mix of pyrrolic and pyridinic Fe coordination centers?

L151: “. And” grammar

L159/160: “pyridinic-N, graphitic-N” The difference should be explained

L 205: “After durability testing and even” I don’t understand

L 206: “high graphitization degree” How is that related to the oxygen content?

L227: “which is among the best stability” Needs to be compared to published data

L234: “under harsh MEA test conditions” This would also involve AST cycles etc.

Reviewer #3 (Remarks to the Author):

The development of Pt-free ORR electrocatalysts is one of the hottest research topics in fuel cell

community. After decades of research, the activity of Pt-free catalysts is approaching to that of Pt. However, the stability, especially the long-term stability under MEA test remains poor, which seriously inhibits the practical application of Pt-free ORR electrocatalysts in fuel cells. In this work, the authors addressed this issue by constructing a monosymmetric Fe-N₄ site, where Fe was coordinated with two pyridinic N and two pyrrolic N atoms. The concept is novel. Besides, the as-prepared catalyst displayed remarkable stability, sustaining stable performance for a duration exceeding 248 h in a fuel cell system. This will provide new insights into develop high-performance and durable Pt-free ORR electrocatalysts. Overall, it is a nice work and can be considered for publication with the following issues addressed.

- i) The authors claimed the monosymmetric Fe-N₄ active site formed by chemical vapor modification by performing EXAFS fitting analysis. More strong evidence is recommended to supplement since the active site structure is important for the stability.
- ii) The other degradation mechanisms, i.e., carbon corrosion, ROS poisoning and water flooding should be excluded or their contribution to the stability should be discussed as well.
- iii) The catalyst structure after durability test in MEA should be supplemented to illustrate the suppressed demetallation of the proposed monosymmetric Fe-N₄ active site.
- iv) Does the chemical deposition modification method completely convert the active site to monosymmetric Fe-N₄ active site and how to control the conversion degree? Is the chemical deposition modification method also suitable to other M-N-C catalysts?

Detailed Response to Reviewers

Reviewer#1

Interesting paper on FeNC electrocatalysts for ORR, tested in acid medium, and air/H₂ PEMFC.

Reply: We deeply appreciate the reviewer for the positive remarks and the suggestive comments to further improve our manuscript. We have carefully addressed all the comments remarked and our point-by-point replies to the comments are shown in detail below.

1. First of all the words “catalyst” must be written as “electrocatalyst” to be precise.

Reply: Thanks for the reviewer’s careful inspection. According to the suggestion, we have replaced all “catalyst” with “electrocatalyst”.

2. Figure 1b/d/c/e (and 1S_all, S10a/b): length bar missing, please add.

Reply: We thank the reviewer for the careful inspection. As pointed out by the reviewer, the length bar has been added in Figure 1b/d/c/e and 1S_all, S10a/b.

The detailed revisions are shown below:

Fig. 1 |. Synthesis and characterization of Fe-N-C and Fe-N-C_{cvm}. a, Schematic

depiction of the synthesis of Fe-N-C_{CVM} electrocatalyst. Scanning electron microscopy (SEM) images of the Fe-N-C (**b**) and Fe-N-C_{CVM} (**c**) electrocatalysts. TEM images of the Fe-N-C (**d**) and Fe-N-C_{CVM} (**e**) electrocatalysts. **f**, Raman spectra of the Fe-N-C and Fe-N-C_{CVM} electrocatalysts. **g**, XRD patterns of the Fe-N-C and Fe-N-C_{CVM} electrocatalysts.

Supplementary Figure 1. SEM images of (a-c) Fe-N-C and (d-e) Fe-N-CCVM.

Supplementary Figure 17. SEM images of the Fe-N-C electrocatalyst (a) and the Fe-

N-CCVM electrocatalyst(b). TEM images of the Fe-N-C electrocatalyst (c) and the Fe-N-CCVM electrocatalyst(d).

3. Figure 2 and related comments on the nature of the active site: the authors demonstrated that the active is $\text{FeN}_2+\text{N}'_2$ in the sample prepared by CVM. How many times the CVM synthesis has been repeated, providing the same $\text{FeN}_2+\text{N}'_2$ results? This is an important point considering the good performance obtained with the FeNC CVM synthesis: repeatability.

Reply:

Thank the reviewers for the constructive comments. The repeatability of the experiment is indeed a very important issue. Due to that the higher stability of $\text{FeN}_2+\text{N}'_2$ than FeN_4 is considered as the major reason for higher stability of the electrocatalysts, we thereby synthesized several batches of the electrocatalysts via our CVM technique and compared them in terms of durability. As depicted in Figure 3c and Supplementary Figure S10, the results indicate that the electrocatalysts produced in three repeated experiments achieved an almost same half-wave potential of 0.8V vs RHE, indicating the excellent repeatability of the experiment. More importantly, we extended the accelerated aging tests to 200,000 and 70,000 cycles, and the electrocatalytic performance of the electrocatalysts from three batches showed no significant decline, providing solid evidence of electrocatalyst repeatability in terms of stability.

Additionally, the X-ray absorption spectroscopy (XAS) technique has been employed to confirm the atomic dispersion of Fe sites in two batches of electrocatalysts and determine the local coordination number. The results indicate a substantial overlap in the Fe K-edges in X-ray absorption near structure (XANES), providing robust evidence once again for the excellent repeatability of Fe-N- C_{CVM} (Figure R1). We then carried out quantitative least-squares EXAFS fitting analysis to obtain the local chelation parameter of the second Fe-N- C_{CVM} . As expected, the fitting results of the Fe-N- C_{CVM} electrocatalyst from the second batch still include two scattering paths, each with a coordination number of 2 (Figure R2), consistent with the same configuration of the electrocatalyst from the first batch ($\text{Fe-N}_2+\text{N}'_2$). The above results conclusively demonstrate the outstanding repeatability of Fe-N- C_{CVM} .

Figure 3c. ORR polarization curves of Fe-N-C_{CVM} before and after different potential cycles between 0.6-1.0V.

Supplementary Figure 10. (a-d) ORR polarization curves of different batches of Fe-N-C_{CVM} before and after different potential cycles between 0.6-1.0V.

Figure R1. Normalized XANES spectra of Fe K-edge for the two batches of Fe-N-C_{CVM}.

Figure R2. Ex situ Fe K-edge Fourier transform EXAFS spectrum and its fitting with the coordination number evenly distributed in the two paths of Fe-N-C_{CVM} from second batch.

4. Figure 3: add in the caption the nature of the electrolyte (0.1M HClO₄) and the mV/s at which each test has been performed. It would be useful to add also the performance of a commercial Pt/C as a reference. The authors claim that the better stability of CVM is due to the presence of FeN₂+N'₂ active sites. In which way exactly these sites make more stable the process? Which reaction mechanism is hypothesized here?

Reply:

Thanks a lot for this helpful advice. According to the reviewer's suggestion, we

added the relevant information regarding the scan rate and the electrolyte in Figure 3. Besides, the ORR polarization curves of Pt/C before and after different potential cycles between 0.6-1.0V has been supplemented in Supplementary Figure 11. The half wave potential of Pt/C decreased by 70 mV after 70,000 cycles of accelerated stress tests. This observation highlights the significantly higher durability of Fe-N-C_{CVM} compared to Pt/C.

We propose that the stability of Fe-N-C_{CVM} can be primarily attributed to the effective inhibition of Fe center atom from leaching, thereby largely preserving the quantity of active sites in the electrocatalyst. Theoretical calculations revealed that FeN₂+N₂ exhibits a much higher Gibbs free energy (23.33 eV) than FeN₄ (8.30 eV) towards metal leaching, suggesting the huge enhancement in structure stability (Figure R3). Moreover, we further conducted post-mortem durability tests by analyzing the electrolyte samples through ICP, where the Fe leaching content of the Fe-N-C_{CVM} electrode is only 12.86 ug/L, substantially lower than 479.39 ug/L for Fe-N-C, as shown in Table R1.

For reaction mechanism, Fe-N-C_{CVM} exhibited a 4-electron reaction pathway. As shown in Figure R4, a Tafel slope of 58 mV dec⁻¹ can be extracted from the LSV curve of Fe-N-C_{CVM}, which is very close to Pt/C (56 mV dec⁻¹). Besides, the H₂O₂ yield is less than 5%, suggesting a nearly complete 4e⁻ ORR pathway (Figure 5a, b).

Fig. 3 | Electrochemical performance of Fe-N-C and Fe-N-C_{CVM} in three-electrode setup. **a**, Cyclic voltammograms of Fe-N-C and Fe-N-C_{CVM} in N₂ saturated 0.1M HClO₄ with a sweep rate at 10 mV/s. **b**, ORR polarization curves of Fe-N-C and Fe-N-C_{CVM} in O₂ saturated 0.1M HClO₄ with a sweep rate at 10 mV/s. ORR polarization curves. **(c)** Fe-N-C_{CVM} and **(d)** Fe-N-C before and after different potential cycles between 0.6-1.0V in O₂ saturated 0.1M HClO₄ with a sweep rate at 10 mV/s. **(e)** electron transfer number and **(f)** H₂O₂ yield of Fe-N-C and Fe-N-C_{CVM}.

Supplementary Figure 10. ORR polarization curves of Pt/C before and after different potential cycles between 0.6-1.0V.

Figure R3. Illustration of (a) FeN_4 evolve into N_4C_{12} and (b) $\text{FeN}_2+\text{N}'_2$ evolve into N_4C_{11} .

Table R1 Fe contents in electrolyte determined by ICP-MS

Samples	Content (ug/L) after 10k cycles	Content (ug/L) after 30k cycles	Content (ug/L) after 50k cycles	Content (ug/L) after 70k cycles
Fe-N- C_{CVM}	7.17	8.94	12.86	13.07
Fe-N-C	176.13	213.69	479.39	

Figure R4. Tafel plots derived from LSV curves for (a) Fe-N-C and Fe-N-C_{CVM}; (b) Pt/C.

5. To discuss about decreasing of TOF without measuring it is a nonsense. Or the authors measure is (TOF and SD, site density), of this is pure speculation. It can be avoided if the measurements cannot be performed. There are several papers that can be used as a reference. A few examples: Appl Catal B Environ 2005, 56:9–35. Nat Commun 2016, 7:1–7. Current Opinion in Electrochemistry 2021, 27:100683.

Reply: Thanks for the reviewer's suggestion. According to the reviewer's suggestion, we have measured the TOF values and incorporated them into the manuscript, along with the addition of references.

For convenience, the changes are shown as follows:

Meanwhile, Fe-N-C exhibits a significant decrease in turnover frequency (TOF) from 0.67 to 0.21 e site⁻¹ s⁻¹ after ASTs (0.8V at pH=1) (Supplementary Figure 13 and Supplementary Figure 14), ascribable to carbon surface oxidation and thereby weakened O₂-binding on iron-based sites. On the contrary, the Fe-N-C_{CVM} sustains its high TOF value of ~ 1.0 e site⁻¹ s⁻¹ (0.8V at pH=1) after ASTs, presumably owing to the overcome of the demetallation problem caused by N protonation (Supplementary Figure 15 and Supplementary Figure 16).³⁶⁻³⁸

- 36 Gasteiger, H. A., et al. Activity benchmarks and requirements for Pt, Pt-alloy, and non-Pt oxygen reduction catalysts for PEMFCs. Appl. Catal. B **56**, 9-35, (2005).
- 37 Malko, D., et al. In situ electrochemical quantification of active sites in Fe-N/C non-precious metal catalysts. Nat Commun **7**, 13285 (2016).
- 38 Zagal, J. H., et al. Mapping transition metal-MN₄ macrocyclic complex catalysts performance for the critical reactivity descriptors. Current Opinion in Electrochemistry **27**, 100683 (2021).

Supplementary Figure 13. (a) Nitrite stripping voltammetry of Fe-N-C in N_2 -saturated 0.5 M acetate electrolyte buffer (pH 5.2) with an electrocatalyst loading of $270 \mu\text{g cm}^{-2}$ and a scan rate of 10 mV s^{-1} . (b) ORR polarization curves of initial Fe-N-C in O_2 -saturated 0.5 M acetate electrolyte buffer (pH 5.2) with an electrocatalyst loading of $270 \mu\text{g cm}^{-2}$ and a scan rate of 10 mV s^{-1} .

Supplementary Figure 14. (a) Nitrite stripping voltammetry of Fe-N-C after ASTs in N_2 -saturated 0.5 M acetate electrolyte buffer (pH 5.2) with an electrocatalyst loading of $270 \mu\text{g cm}^{-2}$ and a scan rate of 10 mV s^{-1} . (b) ORR polarization curves of initial Fe-N-C after ASTs in O_2 -saturated 0.5 M acetate electrolyte buffer (pH 5.2) with an electrocatalyst loading of $270 \mu\text{g cm}^{-2}$ and a scan rate of 10 mV s^{-1} .

Supplementary Figure 15. (a) Nitrite stripping voltammetry of initial Fe-N-C_{CV}M in N₂-saturated 0.5 M acetate electrolyte buffer (pH 5.2) with an electrocatalyst loading of 270 μg cm⁻² and a scan rate of 10 mV s⁻¹. (b) ORR polarization curves of initial Fe-N-C_{CV}M in O₂-saturated 0.5 M acetate electrolyte buffer (pH 5.2) with an electrocatalyst loading of 270 μg cm⁻² and a scan rate of 10 mV s⁻¹.

Supplementary Figure 16. (a) Nitrite stripping voltammetry of Fe-N-C_{CV}M after ASTs in N₂-saturated 0.5 M acetate electrolyte buffer (pH 5.2) with an electrocatalyst loading of 270 μg cm⁻² and a scan rate of 10 mV s⁻¹. (b) ORR polarization curves of Fe-N-C_{CV}M after ASTs in O₂-saturated 0.5 M acetate electrolyte buffer (pH 5.2) with an electrocatalyst loading of 270 μg cm⁻² and a scan rate of 10 mV s⁻¹.

6. Please provide the Fe content before/after stability tests (as done for O and N).

Reply: Thanks a lot for your kind reminder. As suggested by the reviewer, we added the Fe content before and after stability tests in Supplementary Table 5. As shown clearly in the table, the Fe retention is as high as 94.3% for our Fe-N-C_{CV}M electrocatalysts after stability test, far higher than that of the Fe-N-C (68.7%).

Detailed revisions in the supporting information are shown below:

Supplementary Table 5 Fe, N and O contents in electrocatalysts determined by EDS.

Samples	Fe (wt. %)	N (wt. %)	O (wt. %)
Fe-N-C	0.16	3.58	2.09
Fe-N-C after AST	0.11	3.79	6.39
Fe-N-C _{CVM}	0.35	4.62	2.08
Fe-N-C _{CVM} after AST	0.33	4.52	1.74

7. Figure 4a: DOE target values: the one indicated by DOE is 44 mA/cm² @ 900 mV IR-free (and not > 60 at 0.8), as stated in the paper Solid State Ionics 319 (2018) 68–76. Please, refer to the correct one (also by citing this reference), and specify if the value is IR-free or not.

Reply:

Thanks for this constructive suggestion. According to the reviewer's suggestion, we provided the current density at 0.9V and note in the manuscript that the value is iR-free. Also, the reference has been added as Ref 39 (Solid State Ionics 319 (2018) 68–76) in the manuscript. The corresponding changes made are shown below:

Promisingly, Fe-N-C_{CVM} cathode generated 26mA cm⁻² at 0.9 V_{iR-free} (closing to DOE target of 44mA cm⁻² by 2025)³⁹ with peak power density reached 450 mW cm⁻² under H₂ air conditions (Figure 4b), on par with the best PGM-free electrocatalysts (Supplementary Table 6) reported recently.

39 Thompson, S. T., et al. ElectroCat: DOE's approach to PGM-free catalyst and electrode R&D. Solid State Ionics **319**, 68-76 (2018).

Fig. 4 | MEA performance of Fe-N-C and Fe-N-C_{cvm} cathode electrocatalyst. a, Determination of Fe-N-C_{cvm} and Fe-N-C at 0.9V_{iR-free} under 1 bar H₂-O₂. **b,** Polarization and power density curves of Fe-N-C_{cvm} and Fe-N-C. **c,** Polarization and power density curves of Fe-N-C_{cvm} before and after different potential cycles. **d,** Polarization and power density curves of Fe-N-C before and after different potential cycles. **e,** Long-term fuel cell tests for Fe-N-C_{cvm} and Fe-N-C electrocatalysts under H₂-air conditions at a constant potential of 0.67V. Test conditions: cathode loading 4.0 mg cm⁻² for Fe-N-C_{cvm} and Fe-N-C, anode loading 0.1mgPt cm⁻², Nafion 212 membrane, 80 °C, 80 relative humidity (RH) and 1.0 bar H₂-Air at flow rates of 300 ml min⁻¹.

Reviewer #2

In their manuscript entitled “Monosymmetric Fe-N₄ sites enabling super durable PEMFC cathode by chemical vapor modification” the authors report the high durability of a modified FeN₄-type electrocatalyst for ORR. In the chemical vapor

modification (CVM) step the authors claim the transformation of the FeN₄ sites into a FeN₂+N'₂ configuration resulting in a nonsymmetric coordination environment of the atomically dispersed Fe species. While the manuscript provides sufficient data for the modification process of the original substrate by CVM and the newly prepared catalyst shows high durability according to the authors data, publication in Nature Communications cannot be recommended due to the following reasons.

Reply: We thank the reviewer for the suggestive comments to further improve our manuscript. We fully understand the reviewer's skeptical viewpoints on some interpretations, so we have tried our best to reply every question raised in great detail. A large number of supplementary experiments have been conducted for the questions and comments raised, and detailed explanations are shown below.

1. The only noteworthy result is the high catalyst durability

Reply: Thanks for the reviewer's comment. We fully understand reviewer's expectations for new materials and mechanisms. We here would like to explain our new contributions to the community as follows:

a) In this work, we introduced the FeN₂+N'₂ configuration (Figure R5) towards stable operation of nonprecious electrocatalysts in fuel cell, providing a new perspective for developing new M-N-C electrocatalysts.

Thanks to the exceptional durability of this configuration (FeN₂+N'₂), the Fe-N-C_{CVM} electrocatalyst can operate steadily in the MEA for 248 hours. As is widely recognized, the foremost challenge of metal-nitrogen-carbon (M-N-C) electrocatalysts is their insufficient durability leading to rapid deactivation, thus impeding practical applications. Consequently, this work provides a highly valuable reference for the transition of M-N-C towards practical applications.

b) Furthermore, in order to test the reproducibility of our Fe-N-C_{CVM} electrocatalysts, we resynthesized three different batches of electrocatalysts, and tested both their activity and stability. Excitingly, while all three batches exhibited excellent stability after 70k cycles, we even extended the accelerated stress tests to 200k cycles, with no discernible performance deterioration observed (Figure 3c and Supplementary Figure 10). This breakthrough represents a significant advancement in the durability of the electrocatalyst in a three-electrode system. Additionally, to demonstrate that the configuration of our Fe-N-C_{CVM} remains unchanged after performance enhancement, the X-ray absorption spectroscopy (XAS) technique was employed again to confirm the atomic dispersion of Fe sites and determining the local coordination environment (Figure R6). We then carried out quantitative least-squares EXAFS fitting analysis to obtain the local chelation parameter of the Fe-N-C_{CVM-2} (Figure R7). The fitting results of Fe-N-C_{CVM-2} still include two scattering paths, each with a coordination number of 2, consistent with the same configuration of the Fe-N-C_{CVM-1}.

c) The CVM method developed in this study demonstrates excellent universality. We applied the CVM method to Co-N-C and other types of Fe-N_x-C (denoted as Fe-N-C-2). As shown in the Figure R8a, we observed that Co-N-C exhibited a 24 mV decrease in half-wave potential after 50,000 AST cycles. Excitingly, after CVM

treatment, the durability of the electrocatalyst was significantly enhanced, with no apparent decay in half-wave potential after 70,000 AST cycles (Figure R8b). Similarly, Fe-N-C-2, after CVM, did not exhibit noticeable performance degradation in accelerated aging tests (Figure R9). These results fully demonstrate the universality of the CVM method, providing valuable insights for reducing PEMFC costs and promoting the application of non-precious metal electrocatalysts.

Therefore, we believe that we have made important contribution to the M-N-C research community in terms of both understanding and in practical utilization issue: i) understanding the mechanism of forming stable active sites by a CVM technique; ii) pushing the practical usage of Fe-N-C by achieving high stability during a 200k cycles tests, which not only makes a record of the nonprecious based electrocatalysts, but also exceeds that of commercial Pt/C. iii) We have developed a universally applicable strategy to enhance the durability of M-N-C, which is of significant importance for reducing PEMFC costs and promoting its commercial applications.

Figure R5. Illustration of FeN_2+N^2 .

Figure 3c. ORR polarization curves of Fe-N-C_{CVM} before and after different potential cycles between 0.6-1.0V.

Supplementary Figure 10. (a-d) ORR polarization curves of different batches of Fe-N-

C_{CVM} before and after different potential cycles between 0.6-1.0V.

Figure R6. Normalized XANES spectra of Fe K-edge for the two batches of Fe-N-C_{CVM}.

Figure R7. Ex situ Fe K-edge Fourier transform EXAFS spectrum and its fitting with the coordination number evenly distributed in the two paths of Fe-N-C_{CVM}-2.

Figure R8. ORR polarization curves (a) Co-N-C_{CVM} and (b) Co-N-C before and after different potential cycles between 0.6-1.0V in O₂ saturated 0.1M HClO₄ with a sweep rate at 10 mV/s.

Figure R9. ORR polarization curves (a) Fe-N-C-2_{CVM} and (b) Fe-N-C-2 before and after different potential cycles between 0.6-1.0V in O₂ saturated 0.1M HClO₄ with a sweep rate at 10 mV/s.

2.The synthesis method isn't entirely new, CVD is an established method for nanoparticle and surface modification (Zhao, X., Wei, C., Gai, Z. et al. Chemical vapor deposition and its application in surface modification of nanoparticles. Chem. Pap. 74, 767–778 (2020). <https://doi.org/10.1007/s11696-019-00963-y>)

Reply: Thanks for this comment. We fully acknowledge the reviewer's emphasis on innovation. The synthesis method employed in this study is notably different from the traditional CVD.

In traditional CVD, the C and N sources from the upstream, once transported downstream, deposit a layer of material on the surface of electrocatalyst (Figure R10) (Nature energy 2022, 7, 652-663). In contrast, in the developed chemical vapor modification (CVM) in this work, C and N sources transported from upstream etch the surface of the electrocatalyst to remove unstable amorphous carbon and repair

defects in the carbon substrate (Figure R11a). The key difference between CVD and our CVM technique lies in the difference in precursor used between the two. While the traditional CVD technique used ZIF-8, 2-methylimidazole, acetonitrile etc. as the carbon and nitrogen source, we used cyanamide instead, with the etch of unstable C observed in our final sample. SEM and TEM images clearly show the disappearance of amorphous carbon in the electrocatalyst after CVM treatment, with the substrate presents a well-defined and orderly state with distinct edges and corners Figure R11b-g). Moreover, after CVM treatment, the electrocatalyst exhibits a novel configuration of active sites, leading to a leap in the durability of the electrocatalyst. Therefore, CVM represents a novel method for preparing highly stable electrocatalysts.

Figure R10. High-angle annular dark-field IL-STEM images (a, b) of the Fe-AC electrocatalyst at location 2 before and after the CVD process, respectively

Figure R11. (a) Schematic depiction of the synthesis of Fe-N-C_{CVDM} electrocatalyst. Scanning electron microscopy (SEM) images of the Fe-N-C (b, d) and Fe-N-C_{CVDM} (c, e) electrocatalysts. TEM images of the Fe-N-C (f) and Fe-N-C_{CVDM} (g) electrocatalysts.

3. The catalyst ORR performance under PEMFC measurements is reportedly 360 mW cm^{-2} under H_2/air conditions, which to the opposite of the authors claim is not on par with the best PGM-free catalysts, comparing to the values given in Supplementary Table 6 of the manuscript. Fe-AC-AC-CVD reaches 560 mW cm^{-2} , TPI@Z8(SiO₂)-650-C reaches 420 mW cm^{-2} .

Reply: Thanks for this comment. According to the reviewer's suggestions, we have read the mentioned references carefully and optimized the fabrication process of the membrane electrode assemblies to improve the performance of the Fe-N-C_{CVDM} under PEMFC measurements. The specific optimization process is as follows:

a) We optimized the operational environment of MEA. First, we explored the impact of gas flow rates on performance and ultimately determined the appropriate flow rates to be 300 sccm for H_2 and to be 500 sccm Air (Figure R12a). Second, the relative humidity plays a crucial role in the effective establishment of the three-phase interface in the MEA. Therefore, we tested relative humidities of 40%, 60%, 80%, and 100%, with 80% RH showing the best performance (343 mW cm^{-2} Figure R12a-d).

b) The relevant parameters for preparing membrane electrodes were further optimized. We tuned the ink concentration, the compression ratio of MEA, as well as the gas

diffusion layer of the single cell to optimize the fuel cell performance. By increasing the ink concentration (Figure R13), optimizing the compression ratio (Figure R14, 30% compression was found best), replacing H23C2 carbon paper by 3260 (Figure R15), we successfully increased the power density of the single cell to 450 mW cm^{-2} , which is on par with the best PGM-free electrocatalysts (TPI@Z8(SiO₂)-650-C reaches 420 mW cm^{-2}).

In conclusion, through adjustments in gas flow rate, relative humidity, ink concentration, compression ratio, and diffusion layer type, the power density of H₂-Air cell reached 450 mW cm^{-2} , on par with the best PGM-free electrocatalysts (TPI@Z8(SiO₂)-650-C reaches 420 mW cm^{-2}).

Figure R12. (a) Polarization and power density curves of Fe-N-C_{CVM} under different gas flow rates at 80%RH. (b) Polarization and power density curves of Fe-N-C_{CVM} under different gas flow rates at 100%RH. (c) Polarization and power density curves of Fe-N-C_{CVM} at 40%RH. (d) Polarization and power density curves of Fe-N-C_{CVM} at 60%RH. Test conditions: cathode loading 4.0 mg cm^{-2} for Fe-N-C_{CVM}, anode loading $0.1 \text{ mg Pt cm}^{-2}$, $80 \text{ }^\circ\text{C}$ and $1.0 \text{ bar H}_2\text{-Air}$.

Figure R13. Polarization and power density curves of Fe-N-C_{CVM} with different ink concentration. Test conditions: cathode loading 4.0 mg cm⁻² for Fe- N-C_{CVM}, anode loading 0.1mg_{Pt} cm⁻², 80 °C and 1.0 bar H₂-Air.

Figure R14. Polarization and power density Fe-N-C_{CVM} with (a) 20% compression ratio, (b) 30% compression ratio, and (c) 40% compression ratio. Test conditions: cathode loading 4.0 mg cm⁻² for Fe- N-C_{CVM}, anode loading 0.1mg_{Pt} cm⁻², 80% relative humidity, 80 °C and 1.0 bar H₂-Air.

Figure R15. Polarization and power density Fe-N-C_{CVM} with different gas diffusion layers. Test conditions: cathode loading 4.0 mg cm⁻² for Fe- N-C_{CVM}, anode loading

0.1mg_{Pt} cm⁻², 80% relative humidity, 80 °C and 1.0 bar H₂-Air.

4. To be published in Nature Communications I would expect the authors reporting a significant higher power density as for example 700-800 mW cm⁻².

Reply: Thanks for the reviewer's comments. We fully appreciate the reviewers' emphasis on performance. According to the suggestion, we further optimized the performance of MEA, the single cell running under both H₂-Air condition and H₂-O₂ condition were tested.

- 1) On one hand, we optimized the fuel cell performance through optimizing the gas flow rate, relative humidity, ink concentration, compression ratio, and diffusion layer type (Figure R16-18). Electrochemical impedance spectroscopy (EIS) combined with single cell testing results show that 80% relative humidity is conducive to allow for low internal resistance and enhanced cell performance (Figure R16f). Furthermore, a compression ratio of 30% was most conducive to the performance expression of Fe-N-C_{CVM} (Figure R17), along with the choose of more appropriate gas diffusion layer (3260 over H23C2) (Figure R18).
- 2) Through such endeavors, the power density of H₂-Air cell reached 450 mW cm⁻², on par with the best PGM-free electrocatalysts (TPI@Z8(SiO₂)-650-C reaches 420 mW cm⁻²). The H₂-O₂ cell reached 610 mW cm⁻² after such optimizations (Figure R19). While further enhancing the cell performance is possible by further optimizing the electrocatalysts loading, altering the membrane used, and most importantly, increasing the Fe metal loading in the electrocatalyst, we here mainly concentrate on the enhancement of the electrocatalyst stability by our newly developed CVM method.
- 3) We fully acknowledge the reviewers' emphasis on activity. However, for current M-N-C electrocatalysts, the primary challenge hindering their widespread application is the issue of insufficient durability. In the revised manuscript, we further extended the accelerated aging test to 200,000 cycles, with no significant deterioration in the electrocatalyst performance observed, thus successfully breaking the durability record of the electrocatalyst under three-electrode conditions (Figure 3). Furthermore, we assembled Fe-N-C_{CVM} into an MEA cathode for durability test. Specifically, in terms of durability, Fe-N-C_{CVM} demonstrated excellent performance, with only a 4.5% decrease in power density after 20k AST cycles. After 30k cycles, the voltage at 0.8A cm⁻² only decreased by 20mV (from 0.486 V to 0.466 V vs RHE), successfully meeting the DOE 2025 target (≤ 30 mV loss at 0.8 A cm⁻²). More excitingly, 80% of the original power density was retained after 70k AST cycles (Figure 4c) with cell voltage only decrease by 48 mV, indicating robust durability. In contrast, Fe-N-C experienced 40% decrease in power density after 70k accelerated aging test cycles with cell voltage decreased by 171 mV (Figure 4d). This provides valuable insights for reducing the cost of proton exchange membrane fuel cells and promoting the application of non-precious metal electrocatalysts.

Figure R16. Polarization and power density curves of Fe-N-C_{CVM} at (a) dry, (b) 40%RH, (c) 60% RH, (d) 80%RH, (e) 100%RH. Electrochemical impedance spectroscopy (EIS) of Fe-N-C_{CVM} at different relative humidity.

Figure R17. Polarization and power density Fe-N-C_{CVM} with (a) 20% compression ratio, (b) 30% compression ratio, and (c) 40% compression ratio. Test conditions: cathode loading 4.0 mg cm⁻² for Fe-N-C_{CVM}, anode loading 0.1mgPt cm⁻², 80% relative humidity, 80 °C and 1.0 bar H₂-O₂.

Figure R18. Polarization and power density Fe-N-C_{CVM} with different gas diffusion

layers. Test conditions: cathode loading 4.0 mg cm^{-2} for Fe-N-C_{CVM}, anode loading $0.1 \text{ mg Pt cm}^{-2}$, 80% relative humidity, 80 °C and 1.0 bar H₂-O₂.

Figure R19. Polarization and power density curves of Fe-N-C_{CVM}

Fig. 3 | Electrochemical performance of Fe-N-C and Fe-N-C_{CVM} in three-electrode setup. **a**, Cyclic voltammograms of Fe-N-C and Fe-N-C_{CVM} in N₂ saturated 0.1M HClO₄ with a sweep rate at 10 mV/s. **b**, ORR polarization curves of Fe-N-C and Fe-N-C_{CVM} in O₂ saturated 0.1M HClO₄ with a sweep rate at 10 mV/s. ORR polarization curves. **(c)** Fe-N-C_{CVM} and **(d)** Fe-N-C before and after different potential cycles between 0.6-1.0V in O₂ saturated 0.1M HClO₄ with a sweep rate at 10 mV/s. **(e)** electron transfer number and **(f)** H₂O₂ yield of Fe-N-C and Fe-N-C_{CVM}.

Fig. 4 | MEA performance of Fe-N-C and Fe-N-C_{CVM} cathode electrocatalyst. a, Determination of Fe-N-C_{CVM} and Fe-N-C at 0.9V_{iR-free} under 1 bar H₂-O₂. **b,** Polarization and power density curves of Fe-N-C_{CVM} and Fe-N-C. **c,** Polarization and power density curves of Fe-N-C_{CVM} before and after different potential cycles. **d,** Polarization and power density curves of Fe-N-C before and after different potential cycles. **e,** Long-term fuel cell tests for Fe-N-C_{CVM} and Fe-N-C electrocatalysts under H₂-air conditions at a constant potential of 0.67V. Test conditions: cathode loading 4.0 mg cm⁻² for Fe-N-C_{CVM} and Fe-N-C, anode loading 0.1mgPt cm⁻², Nafion 212 membrane, 80 °C, 80 relative humidity (RH) and 1.0 bar H₂-Air at flow rates of 300 ml min⁻¹.

5. Jiao et al. reported a Fe-N-C ORR activity of 33 mA cm⁻² at 0.90 V (iR-corrected; i, current; R, resistance) in a H₂-O₂ proton exchange membrane fuel cell at 1.0 bar and 80 °C. (Jiao, L., Li, J., Richard, L.L. et al. Chemical vapor deposition of Fe-N-C oxygen reduction catalysts with full utilization of dense Fe-N₄ sites. Nat. Mater. 20, 1385–1391 (2021). <https://doi.org/10.1038/s41563-021-01030-2>). The authors should provide similar data for their catalyst under identical conditions, especially the catalyst activity at 0.90 V.

Reply: Thanks for this constructive suggestion. According to the reviewer's suggestion, we have added the current density at $0.9V_{iR-free}$ in our revised manuscript (Figure 4a).

For convenience, the changes are shown as follows:

Promisingly, Fe-N-C_{CVM} cathode generated 130mA cm^{-2} at $0.8 V_{iR-free}$ (Figure 4a) and 26mA cm^{-2} at $0.9 V_{iR-free}$ (closing to DOE target of 44mA cm^{-2} by 2025) with peak power density reached 450 mW cm^{-2} under H₂ air conditions, on par with the best PGM-free electrocatalysts (Supplementary Table 6) recently reported.

Fig. 4 | MEA performance of Fe-N-C and Fe-N-C_{CVM} cathode electrocatalyst. a, Determination of Fe-N-C_{CVM} and Fe-N-C at $0.9V_{iR-free}$ under 1 bar H₂-O₂. **b,** Polarization and power density curves of Fe-N-C_{CVM} and Fe-N-C. **c,** Polarization and power density curves of Fe-N-C_{CVM} before and after different potential cycles. **d,** Polarization and power density curves of Fe-N-C before and after different potential cycles. **e,** Long-term fuel cell tests for Fe-N-C_{CVM} and Fe-N-C electrocatalysts under H₂-air conditions at a constant potential of $0.67V$. Test conditions: cathode loading 4.0 mg cm^{-2} for Fe-N-C_{CVM} and Fe-N-C, anode loading 0.1mgPt cm^{-2} , Nafion 212 membrane, $80\text{ }^{\circ}\text{C}$, 80 relative humidity (RH) and 1.0 bar H₂-Air at flow rates of 300

ml min⁻¹.

6. Most importantly, two of the standard parameters for ORR catalyst activity, the mass activity in A g⁻¹_{catalyst} and Tafel slope in mV dec⁻¹ measured by RDE is missing. This value is essential for catalyst evaluation. For example, Menga et al., report a mass activity of 2.7 ± 0.3 A g⁻¹ and Tafel slope of ≈ 60 mV dec⁻¹ for their Fe-N₄ catalyst containing tetrapyrrolic FeN₄ sites. (Angew. Chem.Int. Ed.2022,61, e20220)

Reply: Thanks a lot for your kind reminder. According to the suggestion, the mass activity in A g⁻¹_{electrocatalyst} and Tafel slope in mV dec⁻¹ measured by RDE have been added to the manuscript.

The detailed revisions are shown below:

In terms of mass activity, Fe-N-C_{CVM} presents a better value of 10 A g_{electrocatalyst}⁻¹ than Fe-N-C (4.17 A g_{electrocatalyst}⁻¹). Additionally, after CVM, the Tafel slope of the electrocatalyst has changed from the original 88 mV dec⁻¹ to 58 mV dec⁻¹ (Supplementary Figure 12). This implies that compared to Fe-N-C, Fe-N-C_{CVM} exhibits a faster kinetic reaction rate.

Supplementary Figure 12. Tafel plots for Fe-N-C_{CVM} and Fe-N-C.

7. Another important parameter for highly porous carbon materials used as electrocatalysts, the apparent surface area as for example measured by BET is missing. A standard carbon support material like Ketjen black™ for example is reported to have 300 or 600 m² g⁻¹. How does the authors catalyst compare to this value or other Fe-N-C based catalysts?

Reply: We thank the reviewer for the careful inspection. According to the suggestion, we have added the BET value in the manuscript.

For convenience, the changes are shown as follows:

While the mesoporous content (2-4 nm) in Fe-N-C_{CVM} significantly increases (Supplementary Figure 3b), the specific surface area decreases from the original 1246 m² g⁻¹ to 1196 m² g⁻¹ after CVM, corroborating the etching of unstable microporous

carbon species to create mesopores.

Supplementary Figure 3. (a) N_2 adsorption/desorption and (b) pore distribution plots of Fe-N-C and Fe-N-C_{CVM}.

8. Some of the figures require revision, as for example figures 4 a and b, where current densities are listed in $A\ cm^{-2}$ and $W\ cm^{-2}$. Figures 3 c and d “1w cycles”, “3w cycles”, and “5w cycles”. Other comments to the authors.

Reply: We appreciate the reviewer for the careful inspection of our manuscript. As suggested by reviewers, we have revised Figure 3 and Figure 4.

The detailed revisions are shown below:

Fig. 3 | Electrocatalytic performance of Fe-N-C and Fe-N-C_{CVM} in three-electrode setup. **a**, Cyclic voltammograms of Fe-N-C and Fe-N-C_{CVM} in N_2 saturated 0.1M $HClO_4$ with a sweep rate at 10 mV/s. **b**, ORR polarization curves of Fe-N-C and Fe-N-C_{CVM} in O_2 saturated 0.1M $HClO_4$ with a sweep rate at 10 mV/s. ORR polarization curves. **(c)** Fe-N-C_{CVM} and **(d)** Fe-N-C before and after different potential cycles between 0.6-1.0V in O_2 saturated 0.1M $HClO_4$ with a sweep rate at 10 mV/s. **(e)** electron transfer number and **(f)** H_2O_2 yield of Fe-N-C and Fe-N-C_{CVM}.

Fig. 4 | MEA performance of Fe-N-C and Fe-N-C_{CVM} cathode electrocatalyst. a, Determination of Fe-N-C_{CVM} and Fe-N-C at 0.9V_{iR}-free under 1 bar H₂-O₂. **b,** Polarization and power density curves of Fe-N-C_{CVM} and Fe-N-C. **c,** Polarization and power density curves of Fe-N-C_{CVM} before and after different potential cycles. **d,** Polarization and power density curves of Fe-N-C before and after different potential cycles. **e,** Long-term fuel cell tests for Fe-N-C_{CVM} and Fe-N-C electrocatalysts under H₂-air conditions at a constant potential of 0.67V. Test conditions: cathode loading 4.0 mg cm⁻² for Fe- N-C_{CVM} and Fe-N-C, anode loading 0.1mgPt cm⁻², Nafion 212 membrane, 80 °C, 80 relative humidity (RH) and 1.0 bar H₂-Air at flow rates of 300 ml min⁻¹.

9. L23: “increased graphitization degree enhancement” What is enhanced and what increased

Reply: Thanks a lot for your kind reminder. As pointed out by the reviewer, we have corrected the error in the mentioned sentence.

The changes in the manuscript are shown as follows:

Herein, by employing the chemical vapor modification method, we successfully drove

the transformation of active sites configuration from FeN₄ to stable monosymmetric FeN₂+N'₂, accompanied by increased graphitization degree in carbon substrate.

10. L46: “high-spin D1 state” Does this refer to ⁵⁷Fe Mössbauer spectroscopy as reported by Wang et al.? (Angew. Chem.Int. Ed.2023,62, e2023042)

Reply: We thank the reviewer for the careful inspection. According to the suggestion, we have added the corresponded references in the manuscript.

For convenience, the changes are shown as follows:

While the Fe atoms in high-spin D1 state are revealed as most active configurations, they suffer rapid deactivation during practical operation^{8,9}.

8. Li, J, et al. Identification of durable and non-durable FeN_x sites in Fe–N–C materials for proton exchange membrane fuel cells. *Nature Catalysis* 4, 10–19 (2021).

9. Liu S, et al. Atomically dispersed iron sites with a nitrogen–carbon coating as highly active and durable oxygen reduction catalysts for fuel cells. *Nature Energy* 7, 652-663 (2022).

11. L97: “mesopores” From SI Figure 3 it appears that 2 different pore sizes exist in Fe-N-C_{CVM}. The authors should specify and discuss this.

Reply: We thank the reviewer for the careful inspection. According to the suggestion, the detailed revisions have been added in the manuscript.

For convenience, the detailed revisions are shown below:

As shown in Supplementary Figure 3a, at higher relative pressures, the occurrence of capillary condensation causes the desorption isotherm to lag above the adsorption isotherm, creating a hysteresis loop. This indicates the presence of mesoporous structures in both electrocatalysts. Interestingly, the hysteresis loop width of Fe-N-C_{CVM} is greater than that of Fe-N-C, suggesting that Fe-N-C_{CVM} has a more abundant mesoporous structure. The pore size distribution chart further validates our speculation. While no significant change in the micropore distribution below 2nm in Fe-N-C_{CVM} is observed in comparison to Fe-N-C, the mesopore (2-4nm) content significantly increases (Supplementary Figure 3b) in the former. The variation in pore structure leads to an overall decrease in specific surface area from the original 1246 m² g⁻¹ to 1196 m² g⁻¹ after CVM, corroborating the etching of unstable microporous carbon species to create mesopores.

Supplementary Figure 3. (a) N₂ adsorption/desorption and (b) pore distribution plots of Fe-N-C and Fe-N-C_{CVM}.

12. L104: “Figure 1d” should be 1g to my understanding

Reply: We appreciate the reviewer for the careful inspection of our manuscript. According to the kind reminder, we have replaced "1d" with "1g" as suggested.

For convenience, the changes are shown as follows:

It is worth noting that no peaks related to metallic crystallographic structures are detected in XRD patterns (Figure 1g), indicating the absence of Fe nanoparticles and clusters after CVM.

13. L116: “Supplementary 5a-h” The Fe distribution is hardly observable

Reply: We appreciate the reviewer for the careful inspection of our manuscript. Following the reviewer's suggestion, we have re-optimized the images to clearly demonstrate the distribution of Fe.

For convenience, the changes are shown as follows:

Supplementary Figure 5. Energy dispersive X-ray spectroscopy (EDS) maps of C (a,e), N (b,f), O (c,g), Fe (d,h) of electrocatalysts after and before chemical vapor etching, respectively.

14. L128: “Figure 2d” Same colors should be used for same species, very confusing

Reply: Thanks a lot for your kind reminder. According to the suggestion, we have represented the same species with the same colors.

For convenience, the changes are shown as follows:

Fig. 2 | The configuration of active sites in the electrocatalysts. a, HAADF-STEM images of the Fe-N-C. **b**, HAADF-STEM images of the Fe-N-C_{CVM}. **c**, Normalized XANES spectra of Fe K-edge for Fe foil, Fe-N-C_{CVM}, Fe-N-C, Fe₂O₃ and FePc. **d**, The k^3 -weighted (k)-function of the EXAFS spectra for Fe foil, Fe-N-C_{CVM}, Fe-N-C and FePc. **e**, EXAFS fitting results of Fe foil, Fe-N-C_{CVM}, Fe-N-C. **f**, N K-edge XANES spectra of Fe-N-C_{CVM} and Fe-N-C. Wavelet transform of Fe K-edge EXAFS data for **(g)** Fe-N-C_{CVM}, **(h)** Fe-N-C and **(i)** Fe foil.

15. L139: “shrinking in Fe-N bond length” Does this reflect a mix of pyrrolic and pyridinic Fe coordination centers?

Reply: We appreciate the reviewer for the careful inspection of our manuscript. The N atoms bonded to Fe can be divided into pyridinic N and pyrrolic N (Li, J, et al. Identification of durable and non-durable FeN_x sites in Fe-N-C materials for proton exchange membrane fuel cells. Nature Catalysis 4, 10–19 (2021)). The binding energy of pyridinic N is lower than that of pyrrolic N, indicating that pyridinic N receives more electrons from Fe. Therefore, the bond length of Fe-pyridinic N is shorter than that of Fe-pyrrolic N. EXAFS reveals that the Fe-N bond length in Fe-N-C_{CVM} is significantly shorter than in Fe-N-C, indicating a change in the local coordination environment of Fe atoms after the CVM process (Figure R20a).

We also carried out quantitative least-squares EXAFS fitting analysis to obtain the local chelation parameter. Interestingly, the best fitting results of Fe-N-C_{CVM} contain two distinct scattering paths, each with a coordination number of 2 (Figure R20b and Supplementary Figure 7), in sharp contrast with a single path best fitting (Supplementary Figure 8) of the Fe-N-C electrocatalyst. It is well known that N coordinated with Fe includes pyridinic N and pyrrolic N. The results provide initial evidence supporting the transformation of the active site configuration in the electrocatalyst from the original FeN₄ to the FeN₂+N'₂ following CVM treatment. Moreover, the type and content of N atoms also contain the information about the type of active site. We therefore employed N XANES spectra and X-ray photoelectron spectroscopic (XPS) N_{1s} spectrum to characterize the type and content of N atoms. In N K-edge XANES spectra, there are two major N 1s → π* at 399.5 eV (d₁) and 401 eV (d₂) corresponding to pyridinic-N and pyrrolic-N (N bonded to two carbon atoms, C-N-C), and one major N 1s → σ* at 408 eV (d₃) corresponding to graphitic N (N bonded to three carbon atoms, N-C₃). As shown in Figure R20c, the Fe-N-C_{CVM} exhibits a much increased d₁ peak intensity and a lower d₂ intensity, indicating the variation in Fe coordination environment after CVM. High-resolution N 1s spectra further corroborate the variation in N structure, with peak intensity associated with pyridinic N (~398.5 eV) increased in the Fe-N-C_{CVM} (Figure R20d). Such alternation is favorable for the transformation in active site configuration, i.e., totally from pyridinic N is highly unlikely, as a large proportion of pyrrolic N still exist in the sample, and it is statistically unrealistic to get a full conversion for all four nitrogen atoms into the pyridinic structure.

Combining all together, we infer that the alternation in Fe scattering path might be associated with the change in Fe coordinated N structure, i.e., the partial transformation from pyrrolic N to pyridinic N and the formation of a FeN₂+N'₂ new structure.

Figure R20. (a) The K3-weighted (k)-function of the EXAFS spectra for Fe foil, Fe-N-C_{CVM}, Fe-N-C and FePc. (b) EXAFS fitting results of Fe foil, Fe-N-C_{CVM}, Fe-N-C. (c) N K-edge XANES spectra of Fe-N-C_{CVM} and Fe-N-C. High-resolution N 1s XPS data of the Fe-N-C and Fe-N-C_{CVM} electrocatalysts.

Supplementary Figure 7. Ex situ Fe K-edge Fourier transform EXAFS spectrum and

its fitting with (a) the coordination number evenly distributed in the two paths, (b) first path filled 3 N atoms, the other path filled 1 N atom, (c) first path filled 1, the other path filled 3 N atoms and (d) four coordination atoms completely filled in a path of Fe-N-C_{CVM}.

Supplementary Figure 8. Ex situ Fe K-edge Fourier transform EXAFS spectrum and its fitting with (a) four coordination atoms completely filled in a path, (b) first path filled 3 N atoms, the other path filled 1 N atom, (c) the coordination number unevenly distributed in the two paths and (d) first path filled 1, the other path filled 3 N atoms of Fe-N-C.

16. L151: “. And” grammar

Thanks for this comment. According to the suggestion, we have corrected this error.

For convenience, the changes are shown as follows:

Atomic scattering path exhibited two different radial distance of Fe-N-C_{CVM} and Fe-N-C, again indicating that the local coordination environment has been changed after chemical vapor modification (Figure 2g, 2h and 2i)

17. L159/160: “pyridinic-N, graphitic-N” The difference should be explained

Reply: Thank for your constructive suggestion. According to the suggestion, the difference between pyridinic-N and graphitic-N have been added to the manuscript.

For convenience, the changes are shown as follows:

In N K-edge XANES spectra, there are two major N 1s → π* at 399.5 eV (d₁) and 401 eV (d₂) corresponding to pyridinic-N and pyrrolic-N (N bonded to two carbon atoms, C-N-C), and one major N 1s → σ* at 408 eV (d₃) corresponding to graphitic N (N

bonded to three carbon atoms, N-C₃).

18. L 205: “After durability testing and even” I don’t understand

Reply: We thank the reviewer for the careful inspection. According to the suggestion, we have corrected this sentence.

For convenience, the changes are shown as follows:

However, the oxygen content of the Fe-N-C_{CVM} electrocatalyst exhibits an opposite trend, with O even decreased from 2.08% to 1.74% after durability test (Supplementary Table 5).

19. L 206: “high graphitization degree” How is that related to the oxygen content?

Reply: We thank the reviewer for the careful inspection. When the carbon substrate undergoes corrosion, its surface inevitably develops hydroxyl, carboxyl, and epoxy groups, leading to an increase in oxygen content in the electrocatalyst. The highly graphitized carbon substrate possesses excellent stability, effectively eliminating carbon corrosion during the ASTs, thereby maintaining its low oxygen content in the electrocatalyst.

20. L227: “which is among the best stability” Needs to be compared to published data

Reply: Thanks for this helpful suggestion. According to the reviewer’s suggestion, we have added additional published data for comparison in Supplementary Table 7.

For convenience, the changes are shown as follows:

Supplementary Table 7 Comparisons of the Long-term fuel cell tests in MEA-level under 1 bar H₂-air in recently published papers.

Electrocatalyst	Operational duration	Test conditions	Reference
0.17/Fe-N-C-kat	170 h	H-Air 0.7V	Angew. Chem. Int. Ed. 2020, 59, 21698-21705 ¹⁴
20Co-NC-1100	100 h	H-Air 0.7V	Adv. Mater. 2018, 30, 1706758 ¹¹
Fe-N-C	25 h	H-Air 0.7V	Adv. Mater. 2019, 31, 1807615 ¹⁵
Co(mIm)-NC	100 h	H-Air 0.7V	Nature Catalysis volume 3, 1044-1054 (2020) ⁹
Fe _S A/FeAC-2DNPC	150 h	H-Air 0.5V	Nat Commun 13, 2963 (2022) ¹⁶
Fe _g -NC/Phen	25 h	H-Air 0.6V	Energy Environ. Sci., 2022, 15, 3033-3040 ¹⁷
Fe-AC-CVD	320 h	H-Air 0.67V	Nature Energy volume 7, 652-663 (2022) ¹²
Fe-NC/Sca	22 h	H-Air	Angew. Chem.Int.

ma-Co-NC

100 h

H-Air 0.7V

21. L234: “under harsh MEA test conditions” This would also involve AST cycles etc.

Reply: Thank you for the constructive suggestions. According to the reviewer’s suggestion, the AST cycles of Fe-N-C and Fe-N-C_{CVM} have been added in Figure 4c and Figure 4d.

The detailed revisions are shown below:

Specifically, in terms of durability, Fe-N-C_{CVM} demonstrated excellent performance, with only a 4.5% decrease in power density after 20k AST cycles. After 30k cycles, the voltage at 0.8A cm⁻² only decreased by 20mV (from 0.486 V to 0.466 V vs RHE), successfully meeting the DOE 2025 target (≤ 30 mV loss at 0.8 A cm⁻²). More excitingly, 80% of the original power density was retained after 70k AST cycles (Figure 4c) with cell voltage only decrease by 48 mV, indicating robust durability. In contrast, Fe-N-C experienced 40% decrease in power density after 70k accelerated aging test cycles with cell voltage decreased by 171 mV (Figure 4d). Such distinctive variation in cell durability further corroborates the much enhanced stability of the electrocatalysts.

Fig. 4 | MEA performance of Fe-N-C and Fe-N-C_{CVM} cathode electrocatalyst. a,

Determination of Fe-N-C_{CVM} and Fe-N-C at 0.9V_{iR}-free under 1 bar H₂-O₂. **b**, Polarization and power density curves of Fe-N-C_{CVM} and Fe-N-C. **c**, Polarization and power density curves of Fe-N-C_{CVM} before and after different potential cycles. **d**, Polarization and power density curves of Fe-N-C before and after different potential cycles. **e**, Long-term fuel cell tests for Fe-N-C_{CVM} and Fe-N-C electrocatalysts under H₂-air conditions at a constant potential of 0.67V. Test conditions: cathode loading 4.0 mg cm⁻² for Fe- N-C_{CVM} and Fe-N-C, anode loading 0.1mgPt cm⁻², Nafion 212 membrane, 80 °C, 80 relative humidity (RH) and 1.0 bar H₂-Air at flow rates of 300 ml min⁻¹.

Reviewer #3

The development of Pt-free ORR electrocatalysts is one of the hottest research topics in fuel cell community. After decades of research, the activity of Pt-free catalysts is approaching to that of Pt. However, the stability, especially the long-term stability under MEA test remains poor, which seriously inhibits the practical application of Pt-free ORR electrocatalysts in fuel cells. In this work, the authors addressed this issue by constructing a monosymmetric Fe-N₄ site, where Fe was coordinated with two pyridinic N and two pyrrolic N atoms. The concept is novel. Besides, the as-prepared catalyst displayed remarkable stability, sustaining stable performance for a duration exceeding 248 h in a fuel cell system. This will provide new insights into develop high-performance and durable Pt-free ORR electrocatalysts. Overall, it is a nice work and can be considered for publication with the following issues addressed.

Reply: We deeply appreciate the reviewer for the affirmative comments on our work in terms of both novelty and significance, as well as the recommendation for publication in nature communications. We are also thankful for the constructive suggestions provided, where the detailed responses are shown below

i) The authors claimed the monosymmetric Fe-N₄ active site formed by chemical vapor modification by performing EXAFS fitting analysis. More strong evidence is recommended to supplement since the active site structure is important for the stability.

Reply: Thanks for this constructive suggestion.

According to the reviewer's suggestion, the ⁵⁷Fe Mössbauer spectroscopy was employed to explore the electron configuration and atomic coordination environment of Fe nuclei in Fe-N-C_{CVM}. As shown in Figure R21, D1 doublets with lower isomer shifts (δ) and quadrupole splitting (QS) values were assigned to the high-spin S1 moiety (Nature energy 2022, 7, 652-663, Nature Catalysis 2020 3, 1044-1054). However, the new doublets with a QS value of 1.57, situated between D1 and D2, has not been reported before (Nature Catalysis 2020 3, 1044-1054, Nature energy 2022, 7, 652-663.). This may correspond to the FeN₂+N₂' configuration.

Meanwhile, while we found that the FeN₂+N₂' configuration gives the best fitting results for the EXAFS results, such result is further supported by the N XANES spectra and X-ray photoelectron spectroscopic (XPS) N 1s spectrum. Specifically, the increased d1 signal at 399.5 eV corresponding to N 1s $\rightarrow \pi^*$ for pyridinic-N versus the decrease in d2 signal at 401 eV (pyrrolic-N) manifests the alteration in N structure after CVM. Meanwhile, high-resolution N 1s spectra further corroborate the alternation in N structure, with peak intensity associated with pyridinic N (~398.5 eV) increased significantly, as shown in Fe-N-C_{CVM} (Figure R22). Such alternation is favorable for the transformation in active site configuration. Furthermore, DFT calculations revealed that the Fe-N bond length in the Fe-N-C_{CVM} configuration (1.97 Å) is significantly shorter than that in the S1 configuration (2.09 Å), which is highly consistent with the results obtained from EXAFS analysis. By

combining the above evidences together, we believe that the configuration of the active site in the electrocatalyst has transformed into $\text{FeN}_2+\text{N}_2'$.

Figure R21. The ^{57}Fe Mössbauer spectroscopy of the Fe-N-C_{cvm}.

Figure R22. (a) N K-edge XANES spectra of Fe-N-C_{cvm} and Fe-N-C. (b) High-resolution N 1s XPS data of the Fe-N-C and Fe-N-C_{cvm} electrocatalysts.

Table R2 Parameters derived from the fittings of ^{57}Fe Mössbauer spectra.

Component	Isomer shift (nm s^{-1})	Quad Splitting (nm s^{-1})	Width
D1	0.41	1.02	0.4
new	0.16	1.56	0.4
Singlet	0	-	0.26

ii) The other degradation mechanisms, i.e., carbon corrosion, ROS poisoning and

water flooding should be excluded or their contribution to the stability should be discussed as well.

Reply: We appreciate the reviewer for the careful inspection of our manuscript. As suggested by the reviewer, we carried out new experiments to investigate the effects of carbon corrosion, ROS poisoning, and water flooding on stability, with discussion shown in below:

- For carbon corrosion, we conducted in situ differential electrochemical mass spectroscopy (DEMS) to compare the initial oxidation potentials of Fe-N-C and Fe-N-C_{CVM}. As shown in Figure R23, the initial oxidation potential of Fe-N-C is 0.71V (Figure R23a), significantly lower than that of Fe-N-C_{CVM} (0.86V). Due to the modifying effect of the CVM process, the graphitization degree of the carbon substrate in Fe-N-C_{CVM} is greatly improved. This improvement in graphitization contributes to the mitigation of carbon corrosion in Fe-N-C_{CVM}, thereby contributing significantly to the stability of the electrocatalyst. The relevant discussions have been added to the manuscript.
- To assess the ROS production level, UV/Vis absorption spectroscopy was employed with 2,20-azinobis(3-ethylbenzthiazoline-6-sulfonate) (ABTS) as substrate. As a widely adopted probe, ABTS can be oxidized by ROS (Figure R24a) and result in a change in the absorbance at 417 nm. As shown in Figure R24b, the absorbance value of Fe-N-C_{CVM} is 80% that of Fe-N-C, which does not show a significant deviation from Fe-N-C, in contrast to the noticeable improvement in durability reported in other literature due to the reduction of Fenton reactions (Figure R25) (Angew. Chem. 2019, 131, 12599-12605). Such a minor difference is not enough to cause a major breakthrough in durability. Therefore, we believe that the reduction in H₂O₂ production in Fe-N-C_{CVM} compared to Fe-N-C is not the primary reason for the much enhanced stability of our Fe-N-C_{CVM} electrocatalyst.
- For water flooding, Fe-N-C_{CVM} and Fe-N-C were assembled as the cathode in an MEA for durability testing under different relative humidity conditions. The results show that Fe-N-C_{CVM} exhibits excellent durability at 60%, 80%, and 100% relative humidity. Especially at the highest humidity, where water flooding is most likely to influence the cell performance, the power density still maintains at 90% after 70k AST cycles (Figure R26). Therefore, we note that the high stability of the cell can be ascribed to the retaining of the active site structure after durability test, and that water flooding is not a major problem for electrocatalysts performance drop after AST cycles.

Figure R23. DEMS signals of CO_2 from the reaction products for (a) Fe-N-C and (b) Fe-N-C_{CVM} in 0.1M HClO_4 .

Figure R24. (a) Reaction between ROS and ABTS; (b) UV/Vis absorption spectra of 0.1M HClO_4 solutions of only ABTS, Fe-N-C_{CVM} and ABTS, Fe-N-C and ABTS, H_2O_2 and ABTS, Fe-N-C_{CVM}, H_2O_2 and ABTS, Fe-N-C as well as H_2O_2 and ABTS.

Figure R25. UV/Vis absorption spectra of 0.1m HClO₄ solutions of only ABTS, Cr/N/C-950 and ABTS, H₂O₂ and ABTS, Fe/N/C-950 and ABTS, Cr/N/C-950, H₂O₂ and ABTS, Fe/N/C-950, as well as H₂O₂ and ABTS after 7 min reaction.

Figure R26. (a) Polarization and power density curves of Fe-N-C_{CVM} at 60%, 80%, and 100% relative humidity. (b) Polarization and power density curves of Fe-N-C_{CVM} before and after different potential cycles at 60% RH. (c) Polarization and power density curves of Fe-N-C_{CVM} before and after different potential cycles at 80% RH. (d) Polarization and power density curves of Fe-N-C_{CVM} before and after different potential cycles at 100% RH.

iii) The catalyst structure after durability test in MEA should be supplemented to illustrate the suppressed demetallation of the proposed monosymmetric Fe-N₄ active site.

Reply: Thanks a lot for the reviewer's constructive suggestion. According to the suggestion, we collected the electrocatalyst (Fe-N-C_{CVM-S}) from the MEA after AST test, with X-ray absorption spectroscopy (XAS) data collected and compared with that of the pristine Fe-N-C_{CVM-S} sample (Figure R27). The Fourier transform of the extended X-ray absorption fine structure (FT-EXAFS) spectra show that the dominant scattering path for Fe-N-C_{CVM} and Fe-N-C_{CVM-S} both occurred at approximately 1.43 Å, indicating that the Fe-N bonds in the electrocatalyst did not undergo significant changes after accelerated stress tests (Figure R28). We then carried out quantitative least-squares EXAFS fitting analysis to obtain the local coordination parameter. Interestingly, the optimal fitting results for Fe-N-C_{CVM-S} include two distinct scattering paths, each with a coordination number of 2, consistent with Fe-N-C_{CVM}, indicating that the active site configuration of Fe-N-C_{CVM} did not change after accelerated stress tests (Figure R29 and Table R3). We further employed X-ray photoelectron spectroscopic (XPS) N_{1s} spectrum to characterize the type and content of N atoms before and after the test. As shown in Figure R30, there is no significant change in the content of pyridinic nitrogen and pyrrolic nitrogen in Fe-N-C_{CVM-S} compared to Fe-N-C, once again confirming that the highly stable active site configuration of our electrocatalyst. Meanwhile, the EDS testing results suggest that our electrocatalyst mains a much higher level of Fe content (94.3% vs. 68.7% for Fe-N-C) after the durability test, as shown in Table 5, further corroborating the higher anti-dissolution stability of our electrocatalysts.

Figure R27. Normalized XANES spectra of Fe K-edge for Fe-N-C_{CVM-S}.

Figure R28. The K3-weighted (k)-function of the EXAFS spectra for Fe-N-C_{CVM} after ASTs, Fe-N-C_{CVM}, and Fe-N-C.

Figure R29. Ex situ Fe K-edge Fourier transform EXAFS spectrum and its fitting with (a) the coordination number evenly distributed in the two paths; (b) four coordination atoms completely filled in a path; (c) first path filled 3 N atoms, the other path filled 1 N atom; (d) first path filled 1, the other path filled 3 N atoms of Fe-N-C_{CVM}.

Table R3. Results of the fitting of the FT-EXAFS spectra collected at the Fe-Kedge for Fe-N-C_{CVM}-s (CN: coordination number; R: distance; σ^2 : mean-square disorder; ΔE_0 : energy shift).

path	CN	R (Å)	ΔE_0 (eV)	$\sigma^2 \times 10^{-3}$ (Å ²)	R factor (%)
Fe-N ₂	2	1.95	3.3	4.72	0.009
Fe-N ₂ '	2	2.05	-0.31	10.00	0.009

Figure R30. High-resolution N 1s XPS data of the Fe-N-C_{CVM}-s and Fe-N-C_{CVM} electrocatalysts (a); Fe-N-C_{CVM} and Fe-N-C (b); Fe-N-C_{CVM}-s and Fe-N-C (c).

Supplementary Table 5 Fe, N and O contents in electrocatalysts determined by EDS.

Samples	Fe (wt. %)	N (wt. %)	O (wt. %)
Fe-N-C	0.16	3.58	2.09
Fe-N-C after AST	0.11	3.79	6.39
Fe-N-C _{CVM}	0.35	4.62	2.08
Fe-N-C _{CVM} after AST	0.33	4.52	1.74

iv) Does the chemical deposition modification method completely convert the active site to monosymmetric Fe-N₄ active site and how to control the conversion degree? Is the chemical deposition modification method also suitable to other M-N-C catalysts?

Reply: We sincerely appreciate the constructive suggestion from the reviewer.

➤ For conversion ratio of the active site: Achieving the complete conversion of all active sites on the surface and in the bulk of the electrocatalyst is indeed a challenging task. As is well known, the occurrence of electrocatalytic reactions takes place on the surface of the electrocatalyst. Therefore, the conversion ratio of active sites on the electrocatalyst surface is of utmost concern. Firstly, we employed XANES spectra to characterize the type and content of N atoms with the TEY mode, which is a technique capable of detecting regions approximately 50 nm below the surface of the electrocatalyst, a distance sufficient for ORR reactions to occur. In N K-edge XANES spectra, there are two major N 1s → π^* at 399.5 eV (d1) and 401 eV (d2) corresponding to pyridinic-N and pyrrolic-N, and one major N 1s → σ^* at 408 eV (d3) corresponding to graphitic N. Excitingly, the Fe-N-C_{CVM} exhibits a much increased d1 peak intensity and a lower d2 intensity, demonstrating that chemical vapor modification can achieve a high conversion ratio of active sites on the surface of electrocatalyst (Figure

R31a). Furthermore, the surface of Fe-N-C_{CVM} was subjected to plasma etching for subsequent XPS characterization. High-resolution N 1s spectra indicated that the content of pyridinic nitrogen and pyrrolic nitrogen on the electrocatalyst after etching is not significantly different from that before etching (Figure R31b), once again confirming the conversion of the configuration of active sites on the surface of the electrocatalyst.

- For application to other M-N-C electrocatalysts: Firstly, we employed a one-pot synthesis to produce Co-N-C electrocatalyst and subjected it to chemical deposition modification. As shown in the Figure R32a, we observed that Co-N-C exhibited a 24 mV decrease in half-wave potential after 50,000 AST cycles. Excitingly, after CVM treatment, the durability of the electrocatalyst was significantly enhanced, with no apparent decay in half-wave potential after 70,000 AST cycles (Figure R32b). Additionally, we synthesized another Fe-N_x-C electrocatalyst (denoted as Fe-N-C-2). The Fe-N-C-2 experienced a 38 mV decrease in half-wave potential after 10,000 cycles and a 52 mV decrease after 30,000 cycles (Figure R33b). Conversely, after CVM treatment, the durability of Fe-N-C-2 was greatly improved, with no significant decay in half-wave potential after 70,000 cycles (Figure R33a). These results fully demonstrate the universality of the CVM method, providing valuable insights for reducing PEMFC costs and promoting the application of non-precious metal electrocatalysts.

Figure R31. (a) N K-edge XANES spectra of Fe-N-C_{CVM} and Fe-N-C; (b) High-resolution N 1s XPS data of the Fe-N-C_{CVM} and Fe-N-C_{CVM}-etching.

Figure R32. ORR polarization curves (a) Co-N-C_{CVM} and (b) Co-N-C before and after different potential cycles between 0.6-1.0V in O₂ saturated 0.1M HClO₄ with a sweep rate at 10 mV/s.

Figure R33. ORR polarization curves (a) Fe-N-C-2_{CVM} and (b) Fe-N-C-2 before and after different potential cycles between 0.6-1.0V in O₂ saturated 0.1M HClO₄ with a sweep rate at 10 mV/s.

REVIEWER COMMENTS

Reviewer #1 (Remarks to the Author):

accept

Reviewer #2 (Remarks to the Author):

Review attached as PDF file.

Reviewer #3 (Remarks to the Author):

The authors have properly addressed my comments. Therefore, the paper is recommended for publication in the present form.

In their revised manuscript “Monosymmetric Fe-N₄ sites enabling super durable PEMFC cathode by chemical vapor modification” the authors addressed my comments satisfactorily. Electrocatalyst performance under PEMFC conditions has improved according to the new data presented and is now on par with previously reported results. The difference between CVD technique and the CVM technique applied by the author has become clearer now. However, before I can recommend publication of the manuscript in Nature Communications, some remaining questions must be addressed, regarding the DOE target of 44 mA cm⁻² by 2025 and the EXAFS spectra and their fitting.

- Regarding the DOE target of 44 mA cm⁻² by 2025, figure 4 a) shows the current generated under 1 bar H₂-O₂. My question, is this operation condition according to the DOE target specification? It would be helpful if the authors could provide a link showing this more clearly.
- Regarding the EXAFS spectra and their fitting, lines 400/401 in the merged document, the authors must provide more details of their EXAFS analysis procedure. Specifically, what kind of background subtraction and smoothing (if any) was applied to the μ_0 . What was the Δk range / nm⁻¹ for the FT into R-space, what was the ΔR range for the curve fitting in R-space?

My questions are specifically based on the fitting results presented in figures 2 e), R30. b), S7 a, b, c). Figure 2 e) [R30. b)] that to my eye don't show a satisfactory fit of the experimentally obtained EXAFS curve in R-space for Fe-N-C_{CVM} in the 2 - 3 Å range. Additionally, in figure R20 a), the EXAFS curves for Fe-N-C_{CVM} and FePc look very similar in the 2 – 3 Å range and if assuming that “FePc” stands for Iron (II)-phthalocyanine with a Fe-N₄ structure (Fe (II) coordinated to four pyrrolic nitrogen atoms) it would contradict the authors claim that Fe-N-C_{CVM} consists of Fe-N₂-N'₂ after synthesis. Judging from supplementary figure 7 (S7) a, b, c) I also cannot clearly see that Fe-N₂-N'₂ (coordination number evenly distributed in the two paths, a), is the correct interpretation because the fitting results of b) and c) (with one path filled one N, the other path filled 3 N), show a similar (c) or even better fit (b) at a radial distance of ~ 2 Å. Contrarily, supplementary figure 21 a) shows an almost perfect fit of Fe-N-C_{CVM} after AST for the Fe-N₂-N'₂ structure. The latter would indicate that the postulated Fe-N₂-N'₂ structure is a result of the electrochemical treatment (AST) and not initially achieved after synthesis. Possibly a mixed structure consisting of various Fe-N_x-N'_y moieties that further convert under electrochemical treatment into Fe-N₂-N'₂ is the initial product. Therefore, the authors should revise their EXAFS fittings to clearly show the proposed Fe-N₂-N'₂ structure after CVM.

Further questions/comments about the merged PDF:

- L178 Fig. 2g should be 2f
- L208: What is the state-of-the art Pt/C catalyst?
- L226: The figure caption (S13, S14) states a pH of 5.2 for ORR, not pH = 1
- L236-238: DEMS should be shown with the corresponding voltaic curve, the onset of carbon oxidation to CO₂ cannot be clearly seen from the provided curves
- L316: What is the S1 configuration, Fe-N₄?
- L442: Figures R12-R15 state a catalyst loading of 4 mg_{cat} cm⁻², which is correct?

Reviewer #1

Accept.

Reply: We appreciate your recognition of our work and your efforts to help us improve it.

Reviewer #2

In their revised manuscript “Monosymmetric Fe-N₄ sites enabling super durable PEMFC cathode by chemical vapor modification” the authors addressed my comments satisfactorily. Electrocatalyst performance under PEMFC conditions has improved according to the new data presented and is now on par with previously reported results. The difference between CVD technique and the CVM technique applied by the author has become clearer now. However, before I can recommend publication of the manuscript in Nature Communications, some remaining questions must be addressed, regarding the DOE target of 44 mA cm⁻² by 2025 and the EXAFS spectra and their fitting.

Reply: Thanks for the reviewer’s positive feedback and the useful suggestions to further improve our manuscript. We have carefully addressed all the comments remarked and our point-to-point replies to the comments are shown in detail below.

1. Regarding the DOE target of 44 mA cm⁻² by 2025, figure 4a) shows the current generated under 1 bar H₂-O₂. My question, is this operation condition according to the DOE target specification? It would be helpful if the authors could provide a link showing this more clearly.

Reply:

Thanks a lot for your kind reminder. The DOE target for the MA of PGM catalysts (<https://www.energy.gov/eere/vehicles/articles/us-drive-fuel-cell-technical-team-roadmap>) is to achieve 44 mA cm⁻² at 0.9V_{iR-free} by 2025, tested at 80°C H₂/O₂ in MEA with fully humidified gas fed and total outlet pressure of 150 kPa (abs). In our paper, in order to better compare with the results shown in literature (Nature Materials 20,1385-1391, 2021, Nature Catalysis 5, 455-462, 2022, Nature Catalysis 2, 259-268, 2019), we adopted the same testing parameter to the one used in literature, i.e., with the cell tested at 80°C under H₂/O₂ in MEA under 1 bar H₂-O₂. It is noted that while a slight difference in testing condition is used compared with the DOE testing protocol, the result allows us to make better comparison between our catalyst and those shown in literature, where our catalysts demonstrated cell performance on par with the best PGM-free electrocatalysts (Supplementary Table 6). To better interpret the data, we have made modifications in our manuscript, with the changes shown as follows:

Promisingly, Fe-N-C_{CVM} cathode generated 130mA cm⁻² at 0.8 V_{iR-free} (Fig. 4a) and 26mA cm⁻² at 0.9 V_{iR-free} under H₂-O₂ mode. The peak power density reached 450 mW cm⁻² under H₂ air conditions (Fig. 4b), on par with the best PGM-free electrocatalysts (Supplementary Table 6) recently reported.

2. Regarding the EXAFS spectra and their fitting, lines 400/401 in the merged document, the authors must provide more details of their EXAFS analysis procedure. Specifically, what kind of background subtraction and smoothing (if any) was applied to the μ_0 . What was the Δk range/nm⁻¹ for the FT into R-space, what was the ΔR range for the curve fitting in R-space?

My questions are specifically based on the fitting results presented in figures 2 e), R30. b), S7a, b, c). Figure 2 e) [R30. b)] that to my eye don't show a satisfactory fit of the experimentally obtained EXAFS curve in R-space for Fe-N-C_{CVM} in the 2-3 Å range. Additionally, in figure R20 a), the EXAFS curves for Fe-N-C_{CVM} and FePc look very similar in the 2-3 Å range and if assuming that "FePc" stands for Iron (II)-phthalocyanine with a Fe-N₄ structure (Fe (II) coordinated to four pyrrolic nitrogen atoms) it would contradict the authors claim that Fe-N-C_{CVM} consists of Fe-N₂-N'₂ after synthesis. Judging from supplementary figure 7 (S7) a, b, c) I also cannot clearly see that Fe-N₂-N'₂ (coordination number evenly distributed in the two paths, a), is the correct interpretation because the fitting results of b) and c) (with one path filled one N, the other path filled 3 N), show a similar (c) or even better fit (b) at a radial distance of ~ 2 Å. Contrarily, supplementary figure 21 a) shows an almost perfect fit of Fe-N-C_{CVM} after AST for the Fe-N₂-N'₂ structure. The latter would indicate that the postulated Fe-N₂-N'₂ structure is a result of the electrochemical treatment (AST) and not initially achieved after synthesis. Possibly a mixed structure consisting of various Fe-N_x-N'_y moieties that further convert under electrochemical treatment into Fe-N₂-N'₂ is the initial product. Therefore, the authors should revise their EXAFS fittings to clearly show the proposed Fe-N₂-N'₂ structure after CVM.

Reply:

Thanks for this constructive suggestion. The raw data was using the IFEFFIT software package according to the standard data analysis procedures. The spectra were calibrated, averaged, pre-edge background subtracted, and post-edge normalized using the Athena program in the IFEFFIT software package. Using E₀ as the zero point, the pre-edge was defined within the range of -150 eV to -30 eV, with a normalization range of 150 eV to 670 eV. The plotting k-weights is 3. To prevent data distortion, spectral curves were not subjected to smoothing. The Δk range/nm⁻¹ for the FT into R-space of Fe-N-C and Fe-N-C_{CVM} is 3-10Å⁻¹. The ΔR range for the curve fitting in R-space is 1-2.2 Å.

For 2-3Å: In the EXAFS curve, the range of 2-3 Å corresponds to either Fe-Fe bonds within the catalyst or the C atoms bonded with N atoms in the second coordination shell of Fe centers. No characteristic peaks of Fe particles or clusters were observed in the XRD (Figure R1a), and neither Fe particles nor Fe clusters were observed in the HAADF-STEM images (Figure R1b). Furthermore, in the wavelet transform images, no Fe-Fe bonds were detected (Figure R1c). Therefore, we believe that Fe in the electrocatalyst exists as single-atom sites, where the range of 2-3 Å corresponds to the second coordination shell C atoms bonded to N atoms, which does not affect our judgment on the configuration of active sites.

Figure R1. (a) XRD pattern of Fe-N-C_{CVM}. (b) HAADF-STEM image of the Fe-N-C_{CVM}. (c) Wavelet transform of Fe K-edge EXAFS data for Fe-N-C_{CVM}.

For fitting results: Following the advice of the reviewers, we have re-fitted the EXAFS data. As shown in the Supplementary Figure 7, compared to other configurations, FeN₂+N'₂ exhibits better fitting results at radial ~ 2 Å, consistent with the suggestion made by the reviewer regarding alignment with the FeN₂-N'₂ configuration. Furthermore, although similar high overlap in the fitting results was observed for FeN₁+N'₃ and FeN₃+N'₁, for EXAFS fitting, the overlap between the fitting curve and the original curve is not the sole criterion for judgment. Parameters such as mean-square disorder (σ^2), R-factor, amplitude attenuation factor (S_0^2), and energy shift (ΔE_0) are equally important, and all of these parameters must fall within reasonable ranges. As shown in Table R1, both FeN₁+N'₃ and FeN₃+N'₁ fitting results exhibit negative σ^2 , indicating that the configurations of FeN₁+N'₃ and FeN₃+N'₁ are not reasonable. Therefore, we believe that the configuration of the active site of the catalyst is FeN₂+N'₂.

Supplementary Figure 7. Ex situ Fe K-edge Fourier transform EXAFS spectrum and its fitting with (a) the coordination number evenly distributed in the two paths, (b) first

path filled 3 N atoms, the other path filled 1 N atom, (c) first path filled 1, the other path filled 3 N atoms and (d) four coordination atoms completely filled in a path of Fe-N-C_{CVM}.

Table R1. Results of the fitting of the FT-EXAFS spectra collected at the Fe-Kedge for Fe-N-C_{CVM} (CN: coordination number; R: distance; σ^2 : mean-square disorder; ΔE_0 : energy shift).

configuration	path	CN	R (Å)	ΔE_0 (eV)	$\sigma^2 \times 10^{-3}$ (Å ²)	R factor (%)
Fe-N ₂ +N' ₂	Fe-N	2	1.87(5)	0.02	2.3	0.021
	Fe-N'	2	2.06	1.51	4.4	0.021
Fe-N ₃ +N' ₁	Fe-N	3	1.91(8)	-0.37	5.1	0.023
	Fe-N'	1	2.10	-6.83	-2.0	0.023
Fe-N ₁ +N' ₃	Fe-N	1	1.86(6)	2.63	-1.0	0.026
	Fe-N'	3	2.01(5)	-1.50	8.7	0.026

3. L178 Fig. 2g should be 2f.

Reply:

We thank the reviewer for the careful inspection. According to the suggestion, we have corrected Fig.2g to 2f.

For convenience, the changes are shown as follows:

As shown in Fig. 2f the Fe-N-C_{CVM} exhibits a much higher d₁ peak intensity and a lower d₂ intensity, indicating the variation in Fe coordination environment after CVM. High-resolution N 1s spectra further corroborate the variation in N structure, with peak intensity associated with pyridinic N (~398.5 eV) increased in the Fe-N-C_{CVM}.

4. L208: What is the state-of-the art Pt/C catalyst?

Reply:

We appreciate the reviewer for the careful inspection of our manuscript. We used a 20 wt.% commercial Pt/C catalyst from Johnson Matthey Company as a comparison sample. According to the kind reminder, we have replaced “the state-of-the art Pt/C catalyst” with “the state-of-the-art commercial Pt/C catalyst”.

For convenience, the changes are shown as follows:

Notably, 200,000 cycles accelerated stress tests (ASTs) induce no performance decay in Fe-N-C_{CVM} (Fig. 3c), which is even better than the state-of-the-art commercial Pt/C catalyst (Johnson Matthey Company) in terms of cycling stability (Supplementary Fig. 10 and Supplementary Fig. 11).

5. L226: The figure caption (S13, S14) states a pH of 5.2 for ORR, not pH = 1

Reply:

Thanks a lot for your kind reminder. The measurement in a pH=5.2 solution was conducted to evaluate the active site density (SD) of the catalysts, while the ORR intrinsic activity was still evaluated in a pH=1 solution. The detailed information is

shown as follows: in order to measure the intrinsic activity and turn over frequency (TOF) of our catalysts, we need to first evaluate the number of active sites, i.e., the site density (SD) of our catalysts. The measurement has to be carried out via NO poisoning experiment in a pH=5.2 solution (sodium acetate and glacial acetic acid), as pH=1 solution induces no NO adsorption effect. The SD_{mass} of the catalyst is evaluated via Eq.1, as shown in below:

$$SD_{mass}(\text{sites} \cdot \text{g}^{-1}) = \frac{Q[A \cdot \text{V} \cdot \text{cm}^{-2}] \times N_A[\text{atom} \cdot \text{mol}^{-1}]}{n \times W[\text{V} \cdot \text{s}^{-1}] \times F[\text{s} \cdot \text{A} \cdot \text{mol}^{-1}] \times L[\text{g} \cdot \text{cm}^{-2}]} \quad (\text{Eq.1})$$

where Q refers to the shaded area in Supplementary Figure13a; n is the number of electrons associated with the reduction per NO stripping; N_A is Avogadro's constant; F is Faraday constant; W is the scan rate; L is the loading of electrocatalyst on the electrode.

Subsequently, with the ORR polarization curve obtained in 0.1M HClO₄, we calculated the kinetic current density (Eq.2) to determine the TOF value of the electrocatalyst (Eq.3).

$$\frac{1}{i_k} = \frac{1}{i_{0.8V}} - \frac{1}{i_L} \quad (\text{Eq.2})$$

$$TOF(e^{-1} \cdot \text{site}^{-1} \text{s}^{-1}) = \frac{i_k[A \cdot \text{g}^{-1}] \times N_A[\text{site} \cdot \text{mol}^{-1}]}{SD_{mass}[\text{sites} \cdot \text{g}^{-1}] \times F[\text{s} \cdot \text{A} \cdot \text{mol}^{-1}]} \quad (\text{Eq.3})$$

Supplementary Figure 13. (a) Nitrite stripping voltammetry of Fe-N-C in N₂-saturated 0.5 M acetate electrolyte buffer (pH 5.2) with an electrocatalyst loading of 270 $\mu\text{g cm}^{-2}$ and a scan rate of 10 mV s^{-1} . (b) ORR polarization curves of initial Fe-N-C in O₂-saturated 0.5 M acetate electrolyte buffer (pH 5.2) with an electrocatalyst loading of 270 $\mu\text{g cm}^{-2}$ and a scan rate of 10 mV s^{-1} .

1. Jiao, L, et al. Chemical vapour deposition of Fe-N-C oxygen reduction catalysts with full utilization of dense Fe-N₄ sites. Nature material 20, 1385-1391 (2021).

2. Wan, X, et al. Iron atom-cluster interactions increase activity and improve durability in Fe-N-C fuel cells. Nature Communications 13, 2963 (2022).

6.L236-238: DEMS should be shown with the corresponding voltaic curve, the onset of carbon oxidation to CO₂ cannot be clearly seen from the provided curves.

Reply:

Thanks for this helpful suggestion. According to the suggestion, we have organized the voltaic curves as shown below (Figure R2). The carbon oxidation potential for Fe-N-C occurs at 0.7V, while the carbon oxidation potential for Fe-N-C_{CVM} is at approximately 0.87V, consistent with the DEMS results. This clearly indicates that after chemical vapor modification, the corrosion resistance of the carbon substrate in the electrocatalyst is significantly enhanced, contributing greatly to the durability of the electrocatalyst.

Supplementary Figure 17. DEMS signals of CO₂ from the reaction products for (a) Fe-N-C and (b) Fe-N-C_{CVM} in 0.1M HClO₄.

Figure R2. (a) The voltaic curve of Fe-N-C. (b) The voltaic curve of Fe-N-C_{CVM}.

7. L316: What is the S1 configuration, Fe-N₄?

Reply:

We appreciate the reviewer for the careful inspection of our manuscript. The S1 configuration is characterized by an Fe-N₄ site coordinated with four pyrrolic nitrogen atoms (Figure R3) (Nature Catalysis 4, 10-19, 2021, Nature Energy 7, 652-663, 2022). For convenience, the changes are shown as follows:

DFT calculations revealed that the Fe-N bond length in the mix configuration (1.97 Å)

is significantly shorter than that in the S1 configuration (Fe-N₄ site coordinated with four pyrrolic nitrogen atoms, 2.09 Å), which is highly consistent with the results obtained from EXAFS analysis.

Figure R3. Illustration of S1 configuration.

8. L442: Figures R12-R15 state a catalyst loading of 4 mg_{cat} cm⁻², which is correct?

Reply:

We thank the reviewer for the careful inspection. We have optimized the loading of the cathode electrocatalyst in the fuel cell. As shown in the Figure R2, when the electrocatalyst loading is too low, the performance of the MEA is poor due to insufficient active site density (Figure R4a). Conversely, when the electrocatalyst loading is too high, a decline in performance is observed, possibly due to reduced mass transport capacity caused by an excessively thick catalytic layer (Figure R4b, Figure R4c, and Figure R4d). Therefore, based on experimental results and literature reports, we have determined the optimal loading to be 4 mg cm⁻² (Nature Energy 7, 652-663, 2022, Nature Catalysis 6, 1215-1227, 2023)

Figure R4. (a) Polarization and power density curves of Fe-N-C_{CVM} with 2.5 mg cm⁻². (b) Polarization and power density curves of Fe-N-C_{CVM} with 4 mg cm⁻². (c) Polarization and power density curves of Fe-N-C_{CVM} with 5.5 mg cm⁻². (d) Comparison of polarization curves for Fe-N-C_{CVM} loadings of 4 mgcm⁻² and 5.5 mgcm⁻².

Reviewer #3

The authors have properly addressed my comments. Therefore, the paper is recommended for publication in the present form.

Reply: Thank the reviewer's valuable comments and hard work, which is very helpful to the improvement of our work.

REVIEWERS' COMMENTS

Reviewer #2 (Remarks to the Author):

The authors have addressed my comments/questions properly. The manuscript is publishable in its present form if the two following points are addressed. (no further review required)

1) The following part is added either to the Methods part (Physical characterizations) of the main text or the supplemental information (EXAFS evaluation): "The spectra were calibrated, averaged, pre-edge background subtracted, and post-edge normalized using the Athena program in the IFEFFIT software package. Using E0 as the zero point, the pre-edge was defined within the range of -150 eV to -30 eV, with a normalization range of 150 eV to 670 eV. The plotting k-weights is 3. To prevent data distortion, spectral curves were not subjected to smoothing. The Δk range/nm⁻¹ for the FT into R-space of Fe-N-C and Fe-N-CCVM is 3-10Å⁻¹. The ΔR range for the curve fitting in R-space is 1- 2.2 Å."

2) Figure R2. is added to supplementary Figure 17 and shown together

Reviewer #2

The authors have addressed my comments/questions properly. The manuscript is publishable in its present form if the two following points are addressed. (no further review required)

Reply: We appreciate your recognition of our work and your efforts to help us improve it.

1) The following part is added either to the Methods part (Physical characterizations) of the main text or the supplemental information (EXAFS evaluation): “The spectra were calibrated, averaged, pre-edge background subtracted, and post-edge normalized using the Athena program in the IFEFFIT software package. Using E_0 as the zero point, the pre-edge was defined within the range of -150 eV to -30 eV, with a normalization range of 150 eV to 670 eV. The plotting k-weights is 3. To prevent data distortion, spectral curves were not subjected to smoothing. The Δk range/nm⁻¹ for the FT into R-space of Fe-N-C and Fe-N-C_{CVM} is 3-10Å⁻¹. The ΔR range for the curve fitting in R-space is 1-2.2 Å.”

Reply: Thanks a lot for your kind reminder. The above part has been added to the Methods part.

2) Figure R2. is added to supplementary Figure 17 and shown together.

Reply: Thanks for this constructive suggestion. Figure R2 has been added to supplementary Figure 17 and shown together.